# HIV-1 Proviral Transcription and Latency in the New Era

**DOI:** 10.3390/v12050555

**Published:** 2020-05-18

**Authors:** Ashutosh Shukla, Nora-Guadalupe P. Ramirez, Iván D’Orso

**Affiliations:** Department of Microbiology, The University of Texas Southwestern Medical Center, Dallas, TX 75390, USA; Ashutosh.shukla@utsouthwestern.edu (A.S.); Nora-guadalupe.Ramirez@UTSouthwestern.edu (N.-G.P.R.)

**Keywords:** HIV-1, provirus, persistence, cure, integration, transcription, ART, latency, latent reservoir, epigenomics

## Abstract

Three decades of extensive work in the HIV field have revealed key viral and host cell factors controlling proviral transcription. Various models of transcriptional regulation have emerged based on the collective information from in vitro assays and work in both immortalized and primary cell-based models. Here, we provide a recount of the past and current literature, highlight key regulatory aspects, and further describe potential limitations of previous studies. We particularly delve into critical steps of HIV gene expression including the role of the integration site, nucleosome positioning and epigenomics, and the transition from initiation to pausing and pause release. We also discuss open questions in the field concerning the generality of previous regulatory models to the control of HIV transcription in patients under suppressive therapy, including the role of the heterogeneous integration landscape, clonal expansion, and bottlenecks to eradicate viral persistence. Finally, we propose that building upon previous discoveries and improved or yet-to-be discovered technologies will unravel molecular mechanisms of latency establishment and reactivation in a “new era”.

## 1. The Latent Reservoir in the Spotlight

The development of combination anti-retroviral therapy (ART), which targets several steps of the viral life cycle, decreased patient viral titers to below the limit of detection using contemporary methods [1,2]. This remarkable biomedical breakthrough was the tipping point for reducing mortality and extending the lifespan of HIV-infected individuals. However, despite initial hopes of undetectable viremia, the discontinuation of ART quickly led to viral titer rebound [3,4], suggesting the presence of a “latent” yet inducible and replication-competent viral reservoir. Consequently, in efforts to control infection, patients must remain on a lifelong regime of ART to prevent the rebound of plasma viremia.

The best characterized latent reservoir in patients consists of integrated viral DNA in resting memory CD4^+^ T cells [2,5], which are refractory to immune surveillance and current ART regime. Studies over the past decade have revealed the timeline, size, stability, and composition of the latent reservoir in ART treated patients.

### 1.1. Timeline, Size, Stability, and Composition of the Latent Reservoir

Kinetically, the latent reservoir is established within days after infection regardless whether the patient has undergone ART [6]. The stability of the latent reservoir in resting memory CD4^+^ T cells in patients receiving highly suppressive therapy with no detectable viremia for many years has been estimated to have a half-life of ~44 months [7]. Given this long-term stability, it is impossible to cure HIV by waiting for the infected cells to decay over time.

In addition to the stability of the latent reservoir, there have been attempts to estimate its size and composition. Initial studies used PCR-based techniques but were later found to have largely overestimated the reservoir magnitude [8,9,10,11,12], as the majority of latently-infected cells in patients hold replication-incompetent proviruses. This discovery indicated that defective genomes are not the source of the rebound virus upon ceasing ART, although they can otherwise contribute to continued immune activation and exhaustion [10,13].

More recent assays, such as the quantitative viral outgrowth assay (QVOA), measure the latent reservoir by computing the number of latently-infected resting CD4^+^ T cells that produce replication-competent virus after in vitro stimulation typically with strong T-cell agonists [14,15]. With this assay, it was estimated that about one in a million resting CD4^+^ T cells in the blood of ART treated patients can be induced to produce replication-competent virus [7,16,17,18]. Nonetheless, in contrast to PCR detection methods, the QVOA largely underestimates the reservoir size as not all intact proviruses reactivate after one round of immune stimulation, i.e., subsequent rounds of treatment can activate a larger (2–3-fold) population of intact proviruses [8,19,20].

In theory, most intact proviruses are capable of producing replication-competent virus. However, the frequency of cells harboring intact proviruses that can, in theory, be stimulated is ~30 times larger than the actual frequency of cells that are induced in the QVOA assay. The collective evidence suggest ~90% of proviral genomes in resting memory CD4^+^ T cells within individuals on ART are defective, and from the remaining ~10% of intact proviruses [8,19,21] only ~0.1–0.2% can be reactivated, posing the question as to where in the human genome is this small fraction of intact and reactivatable proviruses located, and what are the molecular mechanisms underlying their persistence.

Overall, the scarcity of cells containing intact proviruses has posed limitations for defining the composition and underlying molecular mechanisms regulating these types of proviruses. Recent tools to simultaneously define both integration positions and proviral intactness [22,23] are starting to reveal a clearer picture of the “physiologic” latent reservoir and its heterogeneous mechanism of regulation.

Strikingly, Swanstrom and colleagues reported that the genetic diversity of the latent reservoir largely depends on the timing of initiating ART post-infection (i.e., acute or chronic phase) [24], where ~71% of unique intact proviruses induced from post-ART samples were genetically similar to samples taken shortly before ART initiation. Thus, within the intact latent reservoir, there is a small population of inducible proviruses that can potentially reseed the active viral reservoir if ART is interrupted. Further, to sustain viral lifelong persistence, several recent studies have shown the intact, replication-competent proviruses within populations of resting memory CD4^+^ T cells are maintained by inconspicuous levels of clonal expansion [20,25,26,27,28] which could influence targetable approaches. Sékaly and colleagues described that proviruses are maintained by either homeostatic proliferation (in memory CD4^+^ T cells) or antigen stimulation (in resting central memory CD4^+^ T cells) rather than by viral re-infection [29]. Moreover, it was further suggested that clonal expansion can be driven by cytokine and antigen stimulation without causing antiviral effects or inducing viral production, thereby giving key biological insights suggesting an additional challenge to reverse latency in the clinics [30].

### 1.2. Routes of Latency Establishment

The previous discoveries on the timeline, size, stability, and composition of the latent reservoir opened a field on itself and led to the long-lasting debate regarding how the reservoir is established. Different proposed routes have been made concerning how latency establishment occurs after HIV integrates into the genome of target immune cells (naïve and effector CD4^+^ T cell subsets) [4,5,31,32,33,34]. At present, the most widely accepted model of latency establishment is that HIV enters a latent state when effector CD4^+^ T cells transition to a resting memory state due to cellular relaxation, the so-called effector-memory-transition (EMT) [35,36] (Figure 1A, top panel). We refer to this mechanism as “late” latency because the provirus is initially active and later becomes inactive as a potential consequence of cellular relaxation, thereby suggesting a major contribution of host cell state on viral entry into latency. Alternative, non-mutually exclusive models proposed the direct infection of resting CD4^+^ T cells [37], and naive or effector CD4^+^ T cells in which the proviral genome is not efficiently transcribed (Figure 1A, bottom panel). In the latter case, it is likely certain integration sites promote “early” latency, in which the provirus becomes transcriptionally silent immediately after integration into specific, unfavorable genomic areas, despite the beneficial environment for proviral transcription and replication in effector CD4^+^ T cells (Figure 1A, bottom panel).

In contrast to “late” latency, “early” latency suggests a major contribution of integration landscape, but not cell state, of viral entry into latency. As such, the integration site can influence the provirus in different ways: (1) through a nucleosome position and epigenetic-driven mechanism, and/or (2) through an RNA polymerase II (Pol II) transcription-driven mechanism related to transcriptional interference [38,39]. On this topic, several landmark studies have surveyed the chromatin environment of single-copy integrated proviruses in CD4^+^ T cell lines such as Jurkat (the J-Lat clones), including nucleosome positioning [40,41], DNA methylation [42], and histone post-translational modifications (PTMs) [41,43,44,45,46] to study their impact on the establishment and/or maintenance of latency, thereby providing an initial view of the potential effect of integration landscape to proviral transcription and fate (active vs. latent; inducible vs. not inducible). In the sections below, we expand on these topics to describe key discoveries, potential limitations, and urgently needed ideas for future investigations to underpin their contributions to disease progression in physiologically relevant models and patient samples.

One major paradigm about “late” latency is whether it is simply attributed to cells dropping immune signaling dynamics during EMT (Figure 1A, top panel), thereby reducing transcription-sustaining host cell factors, or whether the virus actively “pushes” cells into the resting state by rewiring host transcriptional programs [32], or a combination of both. Conversely, other theories have argued in favor of a cell-autonomous model establishing latency where the virus itself shuts-off viral transcription without host cell-intrinsic or cell-extrinsic contributions [47], indicating a “free solo” scenario. However, the potential key viral players involved in the proposed mechanisms remain to be discovered. Together, these studies potentially indicate that both viral and host cell factors influence the establishment and maintenance of proviral latency.

Despite all these possible routes of latency establishment, one outstanding question in the field to address is whether there are several mechanisms operating at different levels due: (1) to the site of proviral integration into the human genome (weighting more on the proposed “early” latency model) and/or (2) to alterations in cell state leading to cellular relaxation (weighting more on the proposed “late” latency model) (Figure 1A).

Because of the multiple routes of latency establishment and features influencing proviral transcription, including immune cell state, integration landscape, and complex transcription circuit architecture (Figure 1B), it has become difficult to ascertain their precise contribution to proviral fate. Indeed, the numerous possible variations on the origin of latency at the single-cell level may account for the rising of a heterogenous proviral population (Figure 2), whereby cells contain proviruses with variable degrees of activity (expression and replication). While ART stops viral replication in cells containing active proviruses, ART is unable to cope with the transcriptionally silent proviruses and those replicating at very low levels. Therefore, these proviruses survive therapy and generate a latent reservoir that upon ART cessation results in a rapid rebound of plasma viremia. We thus propose that a “new era” of research in this field will encompass uncoupling the influence of various regulatory features (immune cell state, integration landscape, and transcription circuit architecture) on proviral transcription (Figure 1B). We envision that building upon previous discoveries and improved technologies with the advent of genomics (e.g., single-cell) and deep learning will facilitate unraveling molecular mechanisms of latency establishment and reactivation. Further, this will enable the identification of novel cell targets, which may guide strategies to eliminate persistent reservoirs to end the HIV epidemic [48].

Given all these previous landmark discoveries, here, we provide a recount of the literature to discuss the evolution of the field, highlighting the use of various latency models and studies with patient isolates. At the same time, we will be presenting potential limitations of previous discoveries and discuss new approaches needed to move the field forward in our race to devise an effective HIV cure strategy in the “new era”. Specifically, we discuss the following: the role of the nuclear architecture in integration site selection, the current knowledge on how the integration landscape may influence HIV transcription dynamics and proviral fate, the role of nucleosomes and chromatin states, the functions of host cell machinery in regulating transcription, and finally combine these basic discoveries to better understand the therapeutic challenges faced.

## 2. Contribution of Nuclear Topology Shaping the Integration Landscape, Transcriptional Regulation, and Proviral Fate

Studies over the past two decades have defined how HIV integrates into the human genome and how proviral transcription normally operates [49,50,51,52,53]. It has long been known that HIV preferentially integrates into chromatin accessible sites and within or near transcriptionally active regions/genes [54,55,56], which led to the definition of “recurrent integration genes” (RIGs) (Figure 3A) [57,58]. However, integrated proviruses are detected on every human chromosome, in various chromatin landscapes (euchromatic and heterochromatic), and at different locations (intergenic or intragenic) and orientations (sense, divergent or convergent) respective to human genes and regulatory elements (Figure 3B). Despite these landmark discoveries, the underlying molecular basis for this biased integration pattern has long-remained unclear.

### 2.1. Importance of CD4^+^ T Cell Nuclear Topology for HIV Integration

Landmark work by Lusic and colleagues revealed that HIV integration occurs in the outer shell of the nucleus and in close communication with the nuclear pore complex (NPC) [57], thus highlighting a key role for nuclear topology and RIGs strategic positioning for viral integration (Figure 3A). Further evidenced, through electron microscopy, heterochromatin was visualized at the nuclear periphery and overlapping with lamin-associated domains (LADs), whereas euchromatin is typically associated with the NPC resulting in a nonrandom spatial distribution of chromatin (reviewed in [59]). Interestingly, this NPC proximal region contains a subset of host genes characterized by the presence of active transcription chromatin marks in autologous cells. Furthermore, as expected, the heterochromatic regions in the LADs and other transcriptionally active regions located centrally in the nucleus were strongly disfavored [57], consistent with the idea that HIV prefers chromatin accessible regions potentially located in exposed areas of the nucleus and away from LADs (Figure 3A).

Notably, the spatial proximity between the NPC and actively transcribed genes could explain the relationship between HIV integration sites and the first host genomic DNA that HIV would encounter during the translocation of the viral particle from the cytoplasm to the nucleus as a pre-integration complex. Indeed, as the first barrier to access host chromatin, the 30 nucleoporins (Nup) comprising the NPC have been extensively studied in efforts to understand the mechanisms underlying retrovirus integration. By systematically knocking down the expression of each Nup, Bieniasz and colleagues comprehensively explored their contribution to HIV infection and proposed the idea of potential Nup-dependent “pathways” facilitating HIV access to host chromatin [60], implicating nuclear topology with integration site selection. As this review will not focus on the specifics of retroviral integration per se, other key host cell factors facilitating this process are summarized in excellent recent reviews by colleagues in the field [52,61,62].

### 2.2. CD4^+^ T Cell Nucleus Sub-Compartments and Regulatory Elements

The three-dimensional (3D) organization of chromosomes enables long-range interactions between enhancers and promoters that are critical for building complex gene regulatory networks [63,64]. Interphase chromosomes occupy separate spaces known as nuclear territories [65] and each chromosome is organized into dynamic, non-random structures containing stretches of transcriptionally active compartments interspersed with sections of transcriptionally inactive compartments [66]. As such, the genome is partitioned into contact domains (Compartments A and B) (Figure 3A) segregating into smaller sub-compartments (A1, A2, B1, B2, B3, and B4) that (1) appear located in different nuclear territories, (2) are associated with distinct patterns of histone marks, and (3) show different expression levels [67].

Interestingly, RIGs are clustered in specific spatial compartments (sub-compartments A) of the CD4^+^ T cell nucleus [58], which seem to acquire this location during T cell activation, thus explaining their preferred recognition during the integration process (highlighted in green in Figure 3A). Further work to define the genomic features enriched at HIV integration genomic sites and the relevance of their transcriptional activity revealed that RIGs are typically proximal to super-enhancers (referred to as SE). SE were originally defined as locally grouped clusters of enhancers (defined as H3K27ac, H3K4me3, H3K4me1, and active transcription domains within 12.5-kb of each other) driving high levels of transcription of nearby cell-identity genes (Figure 3A) [68]. These results have thus exposed the importance of spatial compartmentalization of the host cell genome for HIV integration during T cell activation, thereby reinforcing the role of nuclear architecture in the infection process as well as in future proviral fate changes.

Recent developments in genomic technologies have led to rapid advances in the study of the 3D genome organization. In particular, Hi-C has been introduced as a method for identifying higher-order chromatin interactions genome-wide [66]. Ren and colleagues have reported megabase-sized local chromatin interaction domains called topologically associated domains (TADs) (Figure 3A) as structural features of the genome organization [69]. TADs boundaries are enriched for the insulator binding protein CCCTC-binding factor (CTCF) forming “CTCF loops” encompassing active sub-compartments A and to block heterochromatin spreading typically present in sub-compartments B (Figure 3A).

Despite these past discoveries, the contributions of nuclear sub-compartments, TADs, and CTCF loops to HIV proviral transcription and fate remains unknown. Since compartments are dynamic in nature, it is likely that their position and chromatin content will change during EMT, thereby potentially converting an active provirus into a resting one during cellular relaxation (Figure 1A). This is consistent with the idea that latency is an unfortunate consequence of infection during a narrow window after CD4^+^ T cell activation [35].

### 2.3. Integration Site Effects on HIV Proviral Transcription and Fate

Notwithstanding the key role of the nuclear architecture in the integration process, and since the integration landscape is highly diverse in sequence and chromatin structure [70,71,72], it seems logical to speculate that the integration site could contain information influencing the magnitude of HIV transcription, thereby shaping the heterogeneous proviral fate (Figure 2). Supporting this idea, landmark work by Verdin and colleagues on tens of Jurkat CD4^+^ T cell clones containing individual HIV integrants revealed that the integration site controls basal and immune stimulation-dependent transcription [71]. This indicates HIV possibly operates in an integration site- and cell state-dependent manner, consistent with theories of “early” and “late” latency establishment (Figure 1A). By analyzing primary patient samples, Nussenzweig and colleagues described most intact proviruses are maintained in silent regions where the cells are in a quiescent state, while the active clonally expanded proviruses are defective, further confirming HIV proviral fate may be cell state- and integration site-dependent (Figure 1B) [21].

Additionally, it is increasingly evident that integration site profiles partially differ in preference for orientation relative to the gene and functional gene classification (i.e., oncogene, housekeeping) between in vitro acutely infected immortalized cell lines (e.g., HeLa, Jurkat) vs. aviremic patient samples under highly suppressive therapy [25,73]. For example, in patient samples, HIV integrated into introns and in the same orientation of proto-oncogenes (e.g., *BACH2, MKL2, STAT5B*) and also upstream of the transcription start sites (TSS). However, these orientation and location preferences/biases were not observed when comparing to in vitro infected cells which had HIV integrated into both orientations relative to the gene and located throughout the genes [25,73]. These studies suggest potential limitations of using in vitro models of integration establishment for understanding the integration landscape effect on HIV proviral transcription and fate decisions and may also indicate a selective pressure in vivo to keep a certain type of integrants.

Strikingly, Nussenzweig and colleagues described preferred integration in “hotspot” locations at or near Alu sequences/repeats in patient samples [21]. Alu are mobile elements that belong to a class of primate-specific retroelements termed SINEs (short interspersed elements), have a copy number well in excess of 1 million, comprise ~11% of the human genome, and are known to have wide-ranging influences on gene expression (reviewed in [74]). Thus, Alu may function as regulatory elements shaping HIV proviral transcription and fate (Figure 3B) in clonally expanded but deficient proviruses and/or in intact proviruses found in CD4^+^ T cells that remain in a quiescent state, a critical aspect that remains to be determined.

Despite these past discoveries, it remains unknown what genomic features or combination of features control HIV proviral expression and fate. It has been known for years that the human genome provides the underlying code for the correct transcriptional regulation of most biological processes through precise spatial-temporal interactions between *cis*-elements present at promoters and enhancers and sequence-specific factors that recognize them, as well as by the position of genes and regulatory elements in the three-dimensional space and in relation to nuclear territories and sub-compartments (Figure 3A). These regulatory elements could provide “local” (related to the processes of transcription and chromatin accessibility) or “distal” (chromatin communication between human and viral genomes) functions that could influence HIV expression, thereby modulating proviral fate (Figure 3C). Depending on the integration site, proviruses might function as constitutively active or silent transcriptional units, and a fraction of the silent proviruses could be reactivated at different levels by immune stimulation. Given these possibilities, we hypothesize that the integration site contains “instructions” or “integration code” (here referred to as “position effect” hypothesis) in which one or more genomic features located proximally and/or distally from the integration site could shape the organization and activity of proviruses in the context of the human genome (Figure 3C).

### 2.4. Conclusions and Future Directions

Notwithstanding our current knowledge regarding how the HIV transcriptional program operates, we have no clear understanding on the role of the integration landscape on proviral fate. Given the large diversity of the integration landscape as well as the complexity in the regulation of transcription, it is conceivable that a combination of regulatory elements (including the location and orientation of proviruses respective to nearby genes and regulatory elements such as enhancers and mobile elements, chromatin states of the provirus and neighboring genomic domains, and long-range chromatin interactions (Figure 3B)) could define the “integration code.” As such, this major biomedical challenge demands a comprehensive definition of the molecular rules modulating proviral fate before we can even leverage this knowledge in the clinical setting. Studies in the “new era” may require deep, integrated genomic approaches, combined with the interrogation of patient samples and the potential implementation of deep learning models to both predict and test molecular features contributing to proviral expression and persistence [75]. First, human genome codes shaping proviral transcription could be deciphered by combining open-source, large-scale datasets (including epigenetics, transcriptome, and 3D genome architecture) to interrogate the chromatin states, transcription activity landscape, nuclear sub-compartments, TADs, and chromatin loops containing the major regulator CTCF around HIV integration sites in CD4^+^ T cells (Figure 3A). Second, regulatory features in the human genome can be predicted and experimentally tested by implementing deep learning approaches. Third, interrogating patient samples with methods that can simultaneously assess proviral integration positions, intactness, and activity at the single-cell level may provide physiologic relevance. Together, these efforts could then be compatible with clinical decision-making, for personalized genome editing-mediated elimination of intact proviruses.

## 3. The HIV Transcriptional Program and Proviral Fate

### 3.1. Phases of the HIV Transcriptional Program

The HIV transcriptional program is composed of various regulatory phases and is controlled by many host cell and viral activators. First, during normal cell homeostasis, “basal” steady-state transcription keeps a low level of non-productive RNA synthesis, leading to short, immature transcripts (Figure 4A). In this state, the viral encoded activator Tat is not expressed, and HIV does not replicate, thereby promoting latency establishment. In the “host” phase, when cells are exposed to immune stimulation, host transcription factors are activated, leading to an initial low-level boost in proviral transcription. In defective proviruses lacking functional Tat [76] or containing mutations in its binding sites on the trans-activation response (TAR) stem-loop [77], which is formed at the 5′-end of nascent viral pre-mRNAs by Pol II, the host phase leads to a small amount of viral products. However, during productive infections with intact proviruses containing functional Tat-TAR axes, the initial transcriptional boost enables Tat synthesis before the “host” phase turns off. In this case, the “host” phase is rapidly followed by a “viral” phase in which Tat amplifies transcription by more than 100-fold, promoting a positive feedback loop and robust viral replication [78] (Figure 4A).

In the basal state of resting CD4^+^ T cells, most host transcription factors that bind the proviral promoter are in an inhibited state but become activated when infected cells encounter a stimulus from the immune microenvironment. For example, ligands (such as antigens and cytokines) trigger several host cell signaling pathways upon receptor activation, inducing the initial translocation of sequestered, cytoplasmic host factors into the nucleus which subsequently recognize their respective binding element at the viral promoter, driving proviral transcription [79,80]. Conceptually, CD4^+^ T cell stimulation functions in a broadly similar manner, i.e., through multiple signaling pathways inducing several master cellular activators [79,81].

### 3.2. Feedback Loop and Theories of Latency Establishment due to Dysfunctions in the HIV Transcriptional Program

While the normal function of the two phases (host and viral) creates a positive feedback loop facilitating viral replication, their interruption promotes latency. However, it has long remained unclear to what extent the host phase influences the outcome of the viral phase. Using experimental and modeling approaches, D’Orso and colleagues recently created a mathematical model recapitulating the complete HIV transcriptional program (“basal-host-viral”). These studies revealed that the host phase is subject to “transcriptional fragility” (due to host co-factor dysfunction), thereby dampening the feedback loop and latency-reversal potential [82] and adding an extra layer of complexity in the mechanisms establishing latency. Concerning patient-derived cells, this model predicts fluctuations in levels of key host co-factors could affect the host phase and, as a consequence, the magnitude of the Tat feedback.

Notably, variations in host co-factor levels alter the outcome of the host phase and impart heterogeneity in the transcriptional responses [82], thereby influencing the reactivation potential of latent HIV. These studies provided a mechanistic explanation on the importance of the host phase to ensure the virus is readily and robustly activated. Interestingly, Weinberger and colleagues have shown that “stochastic” Tat fluctuations drive entry into latency [83], consistent with the idea that the “threshold” of host phase activation influences Tat function [82]. Thus, it is likely that the system stochasticity arises as a consequence of host phase alterations driving Tat synthesis [82]. According to Weinberger, these stochastic fluctuations in Tat drive phenotypic diversity [83], probably originated from transcriptional bursting from the viral promoter [84] and lack of system bistability [85], ultimately influencing the viral latency decision [83]. Thus, the combined transcription heterogeneity and system stochastic variability present a major obstacle for therapeutic strategies to eliminate the latent reservoir [86].

Transcription heterogeneity has been observed in both latently infected clonal CD4^+^ T cell populations [71,87] and primary resting CD4^+^ T cells [8,88]. Even after potent immune stimulation, reactivation occurred only in a fraction of cells, while latent proviruses remained inactive in the majority of the cells [8,89]. Recently, a single-cell RNA-seq (scRNA-seq) approach revealed heterogeneous responses during latent HIV reactivation in primary systems (CD4^+^ T cells) and proposed a model in which cells transition between two alternate states [90], potentially explaining the variable responses.

Given these theories, and to further distinguish whether proviral latency is dependent or independent of cell state, Weinberger and colleagues used an artificially modulated Tat circuit (where the expression of Tat is fused to a controllable-proteolysis tag). They provided evidence that the Tat positive feedback circuit is “hardwired” and sufficient to establish proviral latency, independent of cell activation state [47], thus contradicting theories in which cell state prevails over cell-autonomous, HIV centric mechanisms.

Supporting the importance of cell state, Siliciano and colleagues showed that latency is an unfortunate consequence of infection of CD4^+^ T cells within a narrow time window after activation in which transcriptional reprogramming during EMT (Figure 1A) renders CD4^+^ T cells permissive for latency [35]. These discoveries are also consistent with recent studies by Karn and colleagues showing that entry of effector CD4^+^ T cell subsets into quiescence forces HIV into latency [32], in which the activity of master regulators of the host phase declines over time through their inactivation and sequestration in the cytoplasm (reviewed in [91]). Collectively, while this review does not pretend to address which model is right and if all models are theoretically possible, it is likely that a combination of cell- and HIV-driven forces contribute to the stochastic nature of the HIV transcriptional program.

### 3.3. Conclusions and Future Directions

Given the multiple routes of latency establishment and features influencing proviral transcription, namely immune cell state, integration landscape, and complex circuit architecture (Figure 1A,B), it has become difficult to ascertain their precise contribution to proviral fate (activation, latency, and reactivation). Collectively, we propose that further studies uncoupling their influence on proviral transcription will unravel molecular mechanisms of latency establishment and reactivation, and allow the identification of novel host targets, which may guide strategies to eliminate the persistent yet inducible latent reservoir.

## 4. HIV Proviral Transcriptional Regulation in Homeostatic Conditions

In this section, we will first review the major components (*cis*-elements and *trans*-acting factors) required for the different phases of the HIV transcriptional program and then focus on its regulation during homeostatic conditions in the absence of immune stimulation (basal phase). In the following section, we will then focus on the factors and regulation of the transcriptional program in response to immune stimulation (host and viral phases contributing to the feedback loop) (Figure 4A).

### 4.1. Cis-Elements in the HIV Proviral Genome

The HIV provirus consists of two LTRs located at the 5′ and 3′ ends of the viral genome, but only the 5′-LTR (634-bp in length) exerts greater control as promoter for viral transcription due to 3′-LTR transcription interference [92,93]. Both LTRs are segmented into the U3, R, and U5 regions, where U3 and U5 are further subdivided based on transcription factor binding sites and their impact on HIV proviral transcription activity (Figure 4B). As such, the 5′-LTR can be divided into four functional regions: (1) core promoter (−78 to −1 nt respective to TSS), (2) enhancer (−105 to −79 nt respective to TSS), (3) modulatory region (−454 to −104 nt respective to TSS), and (4) TAR region (+1 to +60 nt respective to TSS) [94]. The 5′-LTR utilizes the host transcription machinery by recruiting a large number of ubiquitously expressed or cell-type specific factors to control proviral transcription (reviewed in [95,96,97,98]). This plethora of host cell factors comprising both activating and repressive activities influence the temporal fate of the provirus (entry into, or exit from, latency).

The modulatory region is bound by dozens of host cell factors (expressed ubiquitously or in a cell-type specific manner), which will not be introduced here but serve to modulate the activity supported by the main regulatory regions (the core and enhancer elements).

The core promoter comprises a TATA box and three tandem GC-rich binding sites for specificity protein 1 (SP1), which together with the general transcription factor IID (TFIID) subunit the TATA box-binding protein (TBP), control the basal phase of the HIV transcriptional program in the absence of any immune stimulation [99,100]. Binding of TFIID initiates the stepwise assembly of general transcription factors TFIIA, TFIIB, TFIIE, TFIIF, and TFIIH, forming the pre-initiation complex (PIC). While PIC assembly occurs without other factors [101], this step alone does not sufficiently explain the high transcriptional rates observed during activation of the positive feedback loop [102].

In addition to the core promoter, the enhancer carries a tandem repeat of 10-bp nuclear factor (NF)-κB (p50–p65 heterodimer) binding sites [79] that cooperatively interacts with the adjacent SP1 to synergistically activate HIV transcription in the host phase of the program [103,104,105]. Additionally, a homodimer of the nuclear factor of activated T-cells (NFAT) can bind to NF-κB–binding motifs in a mutually exclusive manner to facilitate HIV proviral transcription activation in the host phase [81,106,107].

### 4.2. Transcription Factors Acting on cis-Elements in the HIV Proviral Genome

T cell receptor (TCR) stimulation in immortalized models (Jurkat) induces binding of NF-κB to its cognate *cis*-element in the proviral genome, thereby activating proviral transcription [108]. While mutation of the NF-κB–binding motifs moderately impairs proviral expression in immortalized T cells (measured after single round infection of SupT1, Jurkat, and PM1) [109], binding of NF-κB or NFAT to these *cis*-acting elements are critical for provirus activation in primary systems (CD4^+^ T cells) [106,110,111]. Given the importance of NFAT, NF-κB inhibition does not preclude reactivation of latent proviruses from primary models of latency (CD4^+^ T cells) by NFAT suggesting they function independently of each other through a combinatorial mode of action [112].

Consistent with the key roles of NF-κB and NFAT in activating the host phase of the HIV transcriptional program, cytoplasmic sequestration in the absence of cell stimulation is a mechanism proposed for latency in primary models (resting CD4^+^ T cells) [79,81,106,113,114]. Expectedly, this molecular event restricts their nuclear availability to interact with cognate binding elements at the proviral genome. Conversely, in activated CD4^+^ T cells, their nuclear translocation enables efficient proviral transcription (reviewed in [51,115]). Since both NF-κB and NFAT function in the host phase of the HIV proviral transcription program, the mechanism of activation will be further elaborated in the following section.

Another key *cis*-element is an enhancer sequence on the proviral genome (right upstream the NF-κB/NFAT-binding sites) bound by activator protein 1 (AP-1), which is typically a heterodimer of Jun and Fos, or activating transcription factors/cAMP response element binding proteins (ATF/CREB family) [116]. Remarkably, cooperative interactions between AP-1 and NF-κB results in synergistic activation of the proviral genome in the host phase of the program [117], and are key for controlling the establishment and maintenance of proviral latency in immortalized models of latency (Jurkat) [118].

Apart from the important functions of the above host cell factors for controlling basal transcription activity [119], several other sequence-specific factors bind to the 5′-LTR to either enhance or suppress promoter activity (reviewed in [95,96,97]). There is a growing list of host cell factors which modulate proviral transcription activity in the basal state in resting CD4^+^ T cells and displaying repressor activity such as yin yang 1 (YY-1), AP-4, CBF-1, Blimp-1, Ets-2, and FoxO (reviewed in [114,120]). Direct and combinatorial binding of these factors and their associated cofactors to the proviral genome determines basal transcription activity and sets the threshold for the subsequent phases (host and viral) of the proviral transcriptional program, which are known to be regulated by the integration site as well [71].

### 4.3. Pol II Pausing at the HIV Proviral Genome and Elongation Factors

In addition to host cell factors directly binding the proviral genome to control basal transcription during homeostatic conditions, Pol II transcription activity (Figure 4A) is tightly regulated through either recruitment by DNA-binding factors and/or pausing and pause release by a distinct set of factors. Promoter-proximal pausing of Pol II is a common regulatory mechanism of eukaryotic gene transcription after transcription initiation. Initially discovered in *Drosophila* heat shock genes (*Hsp70*), subsequent genome-wide studies have demonstrated that promoter-proximal pausing is a rate-limiting step on the majority of Pol II transcribed genes [121,122], although different classes of genes are regulated through unique mechanisms.

Analogous to Pol II pausing regulation on host cell protein-coding genes, Pol II pausing has been observed at the proviral promoter in various immortalized (Jurkat, U1, and ACH-2 cells) [123,124] and primary (resting central memory CD4^+^ T cells) [82] models of latency at the population cell level (Figure 4A), consistent with the synthesis of short (<100-nt), non-polyadenylated transcripts in the basal phase of the program (in the absence of immune stimulation and no Tat synthesis) [78,125].

Based on these studies, the pioneer discovery of Pol II pausing [121] prompted an exPLoSion of research to define the underlying molecular mechanisms both at the HIV provirus and host genes. These studies have revealed two multi-subunit complexes (negative elongation factor (NELF) and the DRB sensitivity-inducing factor (DSIF)) involved in various aspects of Pol II pausing regulation and blockage of entry into productive elongation [126,127,128].

The first complex regulating pausing is NELF, which is a four-subunit protein complex, comprising NELF-A, B, C/D, and E or RD [128]. The NELF-E subunit binds to nascent pre-mRNA chains available in the Pol II exit channel [128,129,130] and is required for NELF-mediated transcription inhibition [128,131]. Interestingly, the consensus sequence CUGAGGA(U), for NELF-E binding to nascent RNA is present in the loop region of the TAR RNA structure [130]. These discoveries opened the possibility that NELF expression in the resting state only may be directly involved in the maintenance of proviral latency by restricting Pol II pause release and entry into productive elongation. However, NELF is expressed in both resting and effector cell states and recruited to the proviral promoter (5′-LTR) for maintaining proviral latency in immortalized models of latency (U1, Jurkat) [123,124]. Additionally, NELF knockdown in these models produced an increase of Pol II occupancy throughout the proviral genome [123,124]. This result was unexpected since, based on the NELF-dependent Pol II pausing model, one would have expected that loss of NELF should only induce Pol II pause release with a concomitant synthesis of proviral transcripts. However, given the recent discoveries suggesting an intimate coupling between Pol II pausing and initiation control, in which metazoan genes displaying a high degree of pausing tend to display a lower rate of transcriptional initiation [132,133], it seems reasonable that a reduction of Pol II pausing upon NELF loss induces the recruitment of new Pol II molecules for transcription initiation. Nonetheless, it remains unclear if in this scenario Pol II synthesizes productive, fully mature (correctly processed, spliced, and polyadenylated) HIV transcripts. Furthermore, the constitutive loss of NELF-E could induce programs that indirectly feed into the HIV provirus altering its transcription. Given this possibility, approaches exploiting current tools for acute factor depletion, such as auxin-inducible degrons (AID) [134], dTAG [135], or proteolysis-targeting chimeric molecules (PROTACs) [136] seem to be better suited for interrogating HIV proviral transcription changes upon NELF loss. Remarkably, very recent discoveries implementing these approaches to study NELF functions in host genes revealed that NELF regulates a promoter-proximal step distinct from Pol II pause-release [137], thereby illuminating the importance of using improved tools in the “new era” for honing our understanding of HIV proviral transcriptional regulation.

Besides NELF, the second complex relieving the elongation block at the HIV proviral genome is DSIF, which is a heterodimer of SPT4 and SPT5 subunits [138]. Upon transcription initiation, both SPT4 and SPT5 associate with Pol II downstream of the TSS and remain bound until around the site of termination, largely mirroring Pol II distribution [139,140]. Additionally, SPT5 contacts the nascent RNA chain (>18-nt), as it emerges from the elongation complex and subsequently recruits NELF [141]. Furthermore, SPT5 directly interacts with the capping enzyme and enhances mRNA capping [139,142], thus exemplifying the tight coordination of early transcription and RNA processing steps of HIV pre-mRNAs. To overcome the transcription inhibition caused by the negative elongation factors, host cell and viral activators utilize the positive transcription elongation factor (P-TEFb) complex, which will be discussed in the following section.

### 4.4. Conclusions and Future Directions

Taken together, upon integration into the host genome, the provirus is targeted by a plethora of factors, including the basal machinery, sequence-specific regulators, Pol II, and NELF/DSIF, that regulate proviral transcription in homeostatic conditions (basal phase of the HIV transcriptional program). Most previous discoveries derive from in vitro assays, the use of clonal, immortalized models of latency, and some cross-validation using population-based primary models of latency. Although useful, studies using primary models of latency provide population-based, but not site-integration specific information. Thus, it has long been assumed Pol II pauses at all proviruses in homeostatic conditions (Figure 4A), and in response to cell-extrinsic signals, Pol II undergoes pause release and productive elongation to promote proviral activation. However, it remains unknown whether Pol II pausing is a general regulatory feature of all (defective and intact) proviruses and what is the role of the integration site and chromatin environment on the pausing mechanism. This important gap in knowledge in the field should not be ignored, and current models of HIV transcriptional regulation, based on data from immortalized models of latency, should be reconsidered and/or better integrated with more physiologic models in which the precise role of the site of integration on Pol II pausing is further evaluated.

Given these previous caveats, major conclusions stem from models whereby proviruses are integrated into unique, known positions of the human genome (the case of immortalized, clonal models of latency), or in random, typically unknown positions (the case of ex vivo experiments in primary models of latency). Given these findings, an outstanding point is how generalizable these discoveries are in the context of the heterogeneous proviral integrations discussed above (Figure 2). What is the role of integration landscape in the control of basal transcription and Pol II pausing? Do these previous discoveries illuminate the requirement of specific factors and Pol II recruitment and/or pausing only in the models used or can they be translated to, and thus inform about, the infection with defective or intact proviruses in patients under suppressive therapy? Answering these questions will clarify a long-standing dilemma about how useful the immortalized and primary models are, but also provide a fundamental understanding of HIV proviral latency and persistence, which can enlighten future clinical interventions.

## 5. HIV Proviral Transcriptional Regulation During CD4^+^ T Cell Stimulation

As discussed above, during the basal phase of the HIV transcriptional program (Figure 4A), only a handful of short, viral transcripts are synthesized and the virally encoded Tat activator is not yet made, thus impeding viral replication. CD4^+^ T cell stimulation provides an initial low level “boost” to proviral transcription as a consequence of the binding of host activators (e.g., NF-κB, NFAT) to the proviral genome (host phase), which leads to Tat synthesis. Tat then creates a positive feedback loop that ensures robust proviral transcription during the “viral phase” by relieving a block at the elongation step allowing for Pol II productive elongation and processivity [82,143,144] (Figure 4A).

### 5.1. P-TEFb/7SK snRNP Complex

To overcome the elongation block caused by the negative elongation factors, Tat utilizes P-TEFb, which is composed of CDK9 (the kinase catalytic subunit) and CycT1/T2 (the cyclin regulatory subunit) [145,146]. Tat directly recruits P-TEFb to the TAR element formed at the 5′-end of nascent viral pre-mRNAs [145,146,147], where it phosphorylates NELF-E, causing NELF dissociation from TAR at the proviral genome [124,148]. P-TEFb also extensively phosphorylates an evolutionarily conserved repetitive heptapeptide motif in the C-terminal region of SPT5, dislodging DSIF from the paused complex and converting it into an elongation factor that prevents the premature dissociation of viral RNA from transcription complex at terminator sequences [149,150]. Finally, extensive phosphorylation at Ser2 residues of the heptapeptide repeat of the Carboxy-terminal domain (CTD) of RPB1 (the largest subunit of Pol II) overcomes the inhibition imposed by negative elongation factors and Pol II transitions from the paused state to productive elongation [151,152,153].

P-TEFb exists in a functional equilibrium of inactive and active states [154,155,156,157]. In addition to free P-TEFb, the kinase is mostly found in a catalytic inactive state as part of the 7SK small nuclear ribonucleoprotein (snRNP) complex [155]. The snRNP is composed of 7SK RNA, HEXIM protein (hexamethylene bisacetamide inducible) (HEXIM1 or HEXIM2), which directly binds to CycT1 and inhibits the kinase activity [158,159], MePCE (methyl capping enzyme) [160], and LARP7 (La related protein 7) [160,161,162,163]. MePCE and LARP7 protect 7SK snRNA from nucleolytic digestion by binding to stem-loop structures at the 5′ and 3′ ends of RNA, respectively, thereby forming the ‘core’ 7SK snRNP complex [155].

The functional involvement of P-TEFb/7SK snRNP in the regulation of HIV proviral transcription elongation prompted research in two major areas to define the mechanisms of P-TEFb release from the snRNP for kinase activation and its recruitment to the proviral genome. Several mechanisms of P-TEFb release have been proposed and reviewed recently [164] such as through dephosphorylation of the activating CDK9 T-loop [165], through the action of RNA-binding proteins and RNA helicases; and more recently, by Tat recruitment of a ubiquitin ligase for the non-degradative ubiquitination of kinase inhibitor HEXIM1/2 [166].

While it is known that Tat and P-TEFb are recruited to the HIV promoter by binding the TAR element, unexpectedly, even before TAR is synthesized, Tat and 7SK snRNP–bound P-TEFb were also found to be recruited as pre-assembled complex to the proviral promoter in immortalized models of latency (HeLa), and in in vitro reconstituted DNA template assays [167,168]. As the TAR hairpin emerges on the nascent transcript from a promoter-paused Pol II, Tat competitively displaces the inhibitory subunits, potentially owed to the higher affinity of Tat than HEXIM1/2 for CycT1 [169,170], thereby re-positioning P-TEFb from the 7SK snRNP to TAR [167,168], a step that activates the P-TEFb kinase [171]. These discoveries enlightened the possibility that P-TEFb is also initially recruited before TAR-binding and in a Tat-independent manner. Supporting these early ideas, many factors (BRD4, KAP1, ZASC1, HMGA1) and protein complexes (Super Elongation Complex (SEC)) were identified to interact with P-TEFb and modulate delivery of both active and inactive P-TEFb kinase to the proviral promoter [98,172,173,174,175,176]. For the purpose of this review, we will mainly describe factors that have received more attention including SEC, BRD4, and KAP1 to keep this review article focused.

### 5.2. Super Elongation Complex (SEC)

Although initially thought to be recruited as an isolated elongation factor, proteomic and biochemical studies revealed that P-TEFb is recruited to the HIV proviral genome by Tat as part of a larger complex referred to as the “SEC”, which helps relieve Pol II promoter-proximal pausing [174,175]. The SEC is assembled on highly flexible scaffolding subunits (AFF1 or AFF4) that interact with P-TEFb, transcriptional elongation factors (ELL1 or ELL2), ENL or AF9 and EAF1 through their short hydrophobic domains [174,175,177,178]. Although, Tat-mediated transactivation of HIV proviral transcription requires P-TEFb, the SEC is necessary for maximal Tat-dependent activation [179] potentially owed to more robust kinase activity of P-TEFb as part of the SEC compared to the unbound state [180]. Despite being a minor subset of total SECs in immortalized models of latency (Jurkat), AFF1-SEC appears to be the prevalent form of SEC in facilitating HIV proviral activation and latency-reversal [181]. Interestingly, AFF1 enhances the affinity of Tat for CycT1 [179], thereby promoting SEC assembly at the HIV proviral genome, and not like BRD4, which operates in a Tat-independent manner.

### 5.3. BRD4

BRD4 is a member of the large family of the bromodomain and extra-terminal domain (BET) protein family. BRD4 was found to first “extract” the P-TEFb kinase from the 7SK snRNP complex and then deliver it to the proviral promoter through interactions with the regulatory cyclin subunit of the complex [172,173]. Notably, chemical-based approaches have prompted the discovery of BRD4 inhibitors (e.g., JQ1) as useful clinical and basic research tools [182]. JQ1 mimics acetylated histone tails and thus competes with BRD4 binding to the proviral promoter surrounded by acetylated nucleosomes (Figure 4C). Surprisingly, treatment of immortalized latency models (Jurkat) with JQ1 caused latent HIV reactivation, which is counterintuitive given the activating but not repressing nature of BRD4-histone tail interactions [173,183]. However, later, it was found that JQ1 promoted Tat–P-TEFb binding to AFF1-SEC and Tat-dependent reactivation of HIV [181]. Still, additional work in immortalized (Jurkat) and primary (CD4^+^ T cells) models of latency have shown that JQ1, and other bromodomain inhibitors, can reactivate latent proviruses in cells lacking Tat or having defective TAR sequences [184,185], complicating the original models and potentially suggesting indirect effects due to long-term factor depletion or silencing. Supporting the potential nature of indirect effects, displacement of BRD4 from acetylated chromatin with JQ1 or acute BRD4 depletion with a small-molecule degrader dampened transcription elongation independent of P-TEFb recruitment to host gene promoters [186,187].

It is thus likely that BRD4 functions through its atypical built-in kinase activity to phosphorylate Ser2 residues of the Pol II CTD to control Pol II pause release independently of P-TEFb [188,189]. Due to the seemingly conflicting reports, further investigations are required to deconvolute the intertwined relationship between P-TEFb and BRD4. The later work by Bradner and colleagues, using technologies to acutely degrade BRD4 in the minutes time scale, exemplified the need of applying them to study HIV proviral transcription and fate and urge a reconsideration of models implicating BRD4 in P-TEFb recruitment to the proviral genome.

### 5.4. KAP1

Recent studies, in both immortalized (Jurkat) and primary (resting memory CD4^+^ T cells) models of latency, have suggested a role for the transcriptional regulator Kruüppel-associated box (KRAB)-interacting protein 1 (KAP1) (also known as TRIM28, TIF1β) in proviral transcription. KAP1 promotes P-TEFb recruitment to promoter-proximal regions at the proviral genome containing paused Pol II. This occurs prior to, and after, immune stimulation, as part of the 7SK snRNP complex [82,176], in which the kinase remains in a primed state [190]. Thus, 7SK snRNP not only maintains P-TEFb inactive, but also has a positive role in delivering the kinase for “on site” activation while at the right genomic location [176]. These discoveries were unexpected because KAP1 has been previously implicated in transcriptional repression through epigenetic silencing of genes and retroelements in progenitor and non-committed cells as well as repression of viruses in embryonic stem cells [191,192].

Interestingly, the different phases of the HIV transcriptional program have unique functional requirements. Although KAP1 is critical for activation of the host phase, HIV evolved a minimalist system whereby Tat represents a switch to a “higher gear” bypassing KAP1 in the viral phase to activate transcription. The result is, in response to immune stimulation, KAP1 initially recruits P-TEFb to the proviral promoter to facilitate activation by cellular activators, thereafter, Tat functions in a KAP1-independent manner, directly recruiting the kinase to sustain transcription elongation. Given that the host phase has a strict requirement for KAP1, its genetic depletion affects the positive feedback loop, thus reducing the magnitude of reactivation of a latent virus.

Despite the new host phase requirement of KAP1 in proviral transcription, its recruitment mechanism to the proviral promoter and how KAP1 coordinates transcription activation through interactions with the transcription machinery is still obscured. It remains unclear how the state of the cell and the role of immune stimulation influences KAP1-proviral genome recruitment dynamics. Recently, D’Orso and colleagues [193] have shown that KAP1 binds to pathway-specific host transcription activators and hypo-acetylated H4 to regulate Pol II promoter levels and pause release at host genes. Given these recent findings, forthcoming research is needed to assess how this mechanism operates at the proviral genome, the role of cell state and immune stimulation, as well as integration landscape.

An interesting hypothesis has been recently proposed concerning the potential functional interplay between regulatory modules in which BRD4 and KAP1 are part of larger BEC (BRD4-containing elongation complex) and KEC (KAP1-7SK elongation complex) protein complexes, respectively and co-operate to facilitate Pol II pause release and productive elongation [164]. Intriguingly, in their model, KEC functions as a “pre-elongation complex” which can either exclusively deliver already primed P-TEFb to paused promoters for “on site” activation or by “hand-off” mechanism in which the KEC transfers the kinase to BEC and/or SEC for assembly at the promoters, but this model remains to be fully tested in HIV proviruses.

### 5.5. Conclusions and Future Directions

Taken together, work over the past three decades has revealed the foundational groundwork for our understanding of how HIV transcription is regulated in the absence and presence of Tat. The discovery of many host cell factors important for the latency-reactivation switch has enabled both mechanistic and translational studies. Nonetheless, given most of this work has emanated from the use of in vitro assays and models of latency (immortalized and primary), work in the “new era” will tremendously benefit from the advent of single-cell approaches in both primary models and patient samples to facilitate underpinning the contributions of the multiple regulatory features (Figure 1B) to proviral transcription and fate.

## 6. HIV Proviral Nucleosome Positioning and Epigenomics

Following HIV integration into the host genome, a highly ordered chromatin structure is formed encompassing the proviral genome [40,41]. The chromatin structure formed not only governs the establishment and maintenance of proviral latency but also plays a critical role in transcriptional reactivation and latency reversal by serving as the target of chromatin modulators (activators and repressors). The functions of activators/repressors, Pol II, and elongation factors are coordinated by nucleosomes, which are the smallest unit of chromatin. Not merely “beads-on-a-string,” nucleosomes have an active role in the regulation of gene transcription (reviewed in [194]). Given the known functions nucleosomes play in modulating gene activation/repression through chemical modifications and strategic positioning within gene regulatory elements, it is evident that they also exert a layer of HIV transcriptional control, thereby shaping proviral fate. For the purpose of this review, we will mainly focus on chromatin-remodeling complexes regulating nucleosome positioning at the proviral genome in addition to chromatin-modifying complexes (writers, erasers, and readers) regulating the proviral epigenome (Figure 4C).

### 6.1. Chromatin-Remodeling Complexes and Associated Factors

In the early days, the role of chromatin organization in transcriptional regulation of HIV provirus was studied in immortalized models of latency [ACH-2 (T-cell) and U1 (macrophage)] using Micrococcal nuclease (MNase) digestion followed by enzyme restriction analysis [40], which has low resolution and DNA sequence-cleavage biased. In these models, under “basal” conditions, the 5′ LTR of the integrated provirus appeared to contain an array of five well-positioned nucleosomes (nuc-0 to nuc-4) with two nucleosome-free regions (NFR) spanning −255 to −3 and +141 to +265 bp, respective to TSS (+1 nt) (Figure 4C) [40]. While nuc-0 occupied the beginning of the U3 region and contained several promoter and enhancer elements, nuc-1 was positioned in the R-U5 region and flanked on either side by two NFRs [40]. Further, MNase mapping of the 5′-LTR has shown that nucleosome organization appears to be independent of the integration landscape, which was also confirmed using in vitro nucleosome reconstitution [40,41,123,195]. This nucleosomal arrangement is preserved by virtue of chromatin remodelers [196]. The nuc-1 present downstream of the proviral TSS acts as a barrier for transcription initiation [40] but is rapidly disrupted upon activation in response to a plethora of agonists such as pro-inflammatory cytokines (like TNF, activating NF-κB signaling), phorbol esters (like PMA/TPA, activating protein kinase C (PKC)), and histone deacetylase inhibitors (like trichostatin A (TSA), suberoylanilide hydroxamic acid (SAHA)), which appear to induce hyperacetylation of the integrated proviral genome) [40,41,123,196,197]. Interestingly, nucleosome mapping techniques such as indirect end labeling and MNase protection assays revealed nuc-1 position overlaps with the site of Pol II pausing at the proviral genome in immortalized models of latency (U1, ACH-2, and Jurkat), reiterating nuc-1 role in provirus transcription regulation [40,123]. However, experiments of transcription inhibition with α-amanitin, which interacts with the bridge helix in Pol II, thereby interfering with its translocation along the DNA backbone [198], showed that disruption of nuc-1 is independent of transcription [40].

Although the underlying sequence of nuc-1 is thermodynamically unfavorable for nucleosome formation, the position of nuc-1 in the latent proviral state (in Jurkat models of latency) is maintained by the switch/sucrose non-fermentable (SWI/SNF) chromatin remodeling complex [196], which dynamically counteracts the preferred nucleosome over NFR1 onto the energetically suboptimal nuc-1 position. The human analogs of the ATP-dependent SWI/SNF family of chromatin remodelers are composed of two complexes differing in subunit composition: BAF (SWI/SNF-A), which stands for “BRG1/SMARCA4- or BRM/SMARCA2-associated factors”, and PBAF (SWI/SNF-B), which stands for “Polybromo-associated BAF” [199]. Many functions were attributed to these two SWI/SNF complexes including transcriptional activation and repression (reviewed in [200,201]).

Interestingly, both SWI/SNF complexes temporally occupy the proviral promoter with BAF in the basal phase and PBAF during the viral, Tat-dependent activation phase [196]. These observations are consistent with the idea that BAF is crucial for early stages of HIV proviral transcription [202], and that several SWI/SNF subunits interact with Tat [203,204] to facilitate robust proviral activation and sustain the positive feedback loop [196,205,206,207]. In agreement, targeting of BAF by either RNAi-mediated knock down or by pharmacological inhibition with small molecule inhibitors displaced the BAF complex from nuc-1 at the proviral genome [208] which de-repressed basal HIV proviral genome transcription as measured by expression of LTR-driven GFP reporter and viral products. Given this de-repression was in the absence of any CD4^+^ T cell agonists in both immortalized (Jurkat) and primary models of latency, as well as in aviremic patient samples [196,208], it explains the molecular mechanism underlying latency reversal.

The BAF complex appears to be recruited to the proviral genome to maintain latency by one member of the BET family, specifically the shorter isoform of BRD4 [209]. Upon Tat-dependent transcription activation, the p300 histone acetyltransferase (HAT) acetylates Tat in its Arginine-rich RNA-binding domain (Lys50 and Lys51). This step dislodges Tat from TAR RNA [210,211] to enable recruitment of PBAF (PBAF-specific subunit BAF180) to the HIV proviral promoter [196], dictating a potential switch from BAF to PBAF SWI/SNF complexes regulating the exit from latency. In addition to SWI/SNF, Tat hijacks other chromatin regulatory complexes for either preventing or facilitating nucleosome remodeling for HIV transcription activation to rewire proviral fate decisions [212,213,214,215,216,217,218], some of which will be discussed below.

### 6.2. Histone Chaperones and Chromatin Reassembly Factors

In addition to the chromatin-modifying and chromatin-remodeling complexes required to switch “on” or “off” provirus transcription, chromatin reassembly factors (CRFs) like SPT6, CHD1, FACT, and histone H3 chaperones (ASF1a and HIRA) are involved in transcription interference [39,219], thus promoting viral latency by maintaining a repressive chromatin environment after HIV integration into the introns of highly active genes [39,219].

The FACT (facilitates chromatin transcription) complex, a heterodimer of SPT16 and SSRP1, removes and then reinstates the histone H2A/H2B dimer after Pol II elongation [220,221]. Two reports have proposed seemingly opposite functions for FACT in HIV transcription with disparate consequences for proviral fate. First, upon activation, FACT was shown to be recruited to nuc-1 to promote viral transcription [222]. However, a more recent study reported that FACT promotes proviral latency in both immortalized (Jurkat) and primary (resting primary CD4^+^ T) models of latency by interfering with P-TEFb binding to the proviral promoter [223]. These conflicting reports could be due to elongation factors repressing transcription initiation from cryptic HIV promoters integrated within host coding regions [224] and not solely to FACT obstructing the association between P-TEFb and Tat [223], or to unwanted indirect effects due to long-term FACT loss. As such, to fathom FACT functions in the maintenance of, and/or activation from, latency, future studies should prioritize the use of more consistent physiologic models of latency and acute depletion approaches to eliminate unnecessary variables.

### 6.3. Proviral Chromatin Acetylation/Deacetylation and Associated Factors

Nucleosome positioning is typically concerted with the deposition and/or removal of histone tail PTMs (Figure 4C). For example, the lack of histone tail (primarily H3 and H4) Lys acetylation facilitates interactions between those tails and DNA [225], thus reinforcing the role of DNA sequences in guiding nucleosome positioning.

Histone tails are post-translationally modified to establish regulatory codes that facilitate or prevent factor-chromatin interactions [226]. Specifically, cycles of histone acetylation/deacetylation are required to precisely fine-tune the magnitude and kinetics of transcriptional programs. Given HIV integrates into the host genome, proviral transcription is controlled through enzymes that deposit or remove acetyl groups, namely HATs and histone deacetylases (HDACs), respectively. Thus, recruitment of HDACs to the proviral genome (through transcription factors, like NF-κB/p50) in the basal phase are thought to constantly keep histone (H3 and H4) tails in the unacetylated or hypoacetylated state [46]. Upon transcription activation in the host phase (in response to immune stimulation), HDACs are evicted from the proviral genome and several HATs including CREB-binding protein (CBP), GCN5, and P/CAF are recruited to the proviral genome in immortalized models of latency (U1 and HL3T1 cells) causing the expected increase in histone (H3 and H4) acetylation levels (hyperacetylation) of nucleosomes encompassing the proviral 5′-LTR [213,214,215,216,227]. Nucleosomes upstream and downstream of the TSS remained hypoacetylated in the basal state, but acquired a specific hyperacetylation pattern with increased H3K9ac, H3K14ac, H4K5ac, H4K8ac, and H4K16ac, but not H4K12ac, upon transition to the host-viral phases in response to strong immune stimulation with phorbol esters (PMA) [227].

Since HDACs prevent histone tail acetylation, thereby promoting chromatin compaction, it came to no surprise that HDAC deposition at the proviral genome was linked to, and strongly correlated with, latency maintenance and establishment [41,43]. Thus, expectedly, treatment with HDAC inhibitors (trapoxin (TPX) and TSA, SAHA, romidepsin) triggered a HAT-dependent nuc-1 remodeling, thereby facilitating spontaneous HIV proviral expression in both immortalized (ACH2, OM 10.1, Ul, J49, J-Lat) and primary (resting and memory CD4^+^ T cells) models of latency [41,46,228,229,230,231]. HDAC activity inhibition promoted Pol II recruitment to the proviral genome but remained non-processive generating only short viral transcripts, consistent with the idea of activation of transcription initiation but not elongation [46].

The HDAC family is composed of five classes (I, IIA, IIB, III, and IV) based on function and sequence similarity (reviewed in [232]). Specifically, HDAC1 within class I co-purified with the yin yang 1–late SV40 factor (YY1-LSF) transcription repressor complex, which binds nuc-1 region on the proviral genome [43,233]. As a result, HDAC1 keeps nuc-1 in a hypoacetylated state, potentially reinforcing nucleosome positioning to prevent spurious activation [43,233] (Figure 4B). Despite strong molecular evidences linking HDAC-mediated control of proviral latency, clinical data later suggested that HDAC inhibition alone was insufficient for latency reversal due to low levels of reactivation and loss of effectiveness after prolonged treatment [234,235]. This may be potentially explained by the generation of read-through transcripts from upstream host promoters [236] indicating a lack of fully-mature and processed HIV transcripts.

### 6.4. Proviral Chromatin Methylation/Demethylation and Associated Factors

Besides histone acetylation/deacetylation, histone methylation of nucleosomes, by histone methyltransferases (HMTs), surrounding the proviral genome is another epigenetic mechanism contributing to nucleosome positioning, thereby regulating proviral fate. Multiple HMTs (SUV39H1, SUV39H2, and G9a) target distinct chromatin domains with different degrees of histone tail Lys methylation (e.g., mono-, di-, or tri-methylation) [237]. For instance, mono- and di-methylation of H3K9 (H3K9me1/2) are localized within silent domains of euchromatic regions, while tri-methylation of H3K9 (H3K9me3) is enriched at pericentric heterochromatin [237]. In immortalized models of latency (Jurkat), high levels of H3K9me3 were present at the proviral 5′-LTR, which is rapidly lost after immune stimulation (TNF-α) [238]. The SUV39 family of SET-domain containing HMTs, SUV39H1 and G9a, both appear to associate with the proviral 5′-LTR, which correlates with H3K9me2/3 levels to establish and maintain latency [239,240]. First, SUV39H1 binds to the proviral 5′-LTR in immortalized models of latency (HeLa-LTR-luc) and establishes a transcriptionally repressed state by maintaining high levels of H3K9me3 [239]. Second, G9a is responsible for proviral transcription repression in immortalized models of latency (ACH-2) by promoting H3K9me2 of nucleosomes surrounding the 5′-LTR [240]. Third, the SETDB1 HMT appears to regulate proviral transcriptional activity through Tat methylation [241].

Despite previous evidence that these three HMTs regulate proviral transcription, it is still unclear how they target proviruses integrated in diverse chromatin states. Since HMT have varying mechanisms of action at different chromatin domains [237], it is possible that proviruses integrated into different chromatin contexts may be targeted by different, or combination of, HMTs to promote the establishment of latency. This would be achieved through regulation of histone PTMs and nucleosome positioning enforcement. In this context, the role of the transcriptional repressor heterochromatin protein-1 (HP1) is noteworthy. The HP1 family in human cells is composed of three isoforms (HP1-α, HP1-β, and HP1-γ), which bind directly to methylated H3K9 tail but not to unmodified H3 [242]. Although the three HP1 isoforms share sequence similarity and structural organization, their nuclear localization respective to chromatin territories is different [243]. HP1-α and HP1-β are primarily associated with centromeric heterochromatin (like SUV39H1 and SUV39H2), whereas HP1-γ is found in both heterochromatic and euchromatic regions (like G9a) [244]. Interestingly, HP1-γ is present in the transcribed regions of active genes and physically associate with the elongating form of Pol II [245,246]. Remarkably, experiments in *Xenopus* oocytes showed that HP1 recruitment to chromatin requires not only the presence of H3K9 methylation but also a direct interaction of HP1 with HMTs. Thus, while both SUV39H1 and G9a have H3K9 methyltransferase activity, only SUV39H1 is able to recruit all isoforms of HP1 to chromatin through direct interaction [247].

The multiple HP1 isoforms and their different chromatin distribution profiles opened the question as to which HP1s regulate HIV proviral transcription, and what are the underlying molecular mechanisms. Surprisingly, distinct HP1 isoforms were found to associate with the proviral genome in various models and under different contexts, thus questioning the central mechanism.

In 2007, Benkirane and colleagues ascribed HP1-γ function, but not HP1-α and HP1-β, in maintaining provirus latency in both immortalized (HeLa and Jurkat) models of latency and PBMCs isolated from aviremic patients [239]. In 2008, the Karn lab reported HP1-α occupancy at the proviral promoter in the basal state and its eviction during the transition to the host-viral phases of the program in response to immune stimulation (TNF-α) in immortalized models of latency (Jurkat) [238]. The same year, a third study found HP1-β bound to the proviral genome in the basal state and a switch from HP1-β to HP1-γ during the transition to the host-viral phases of the program in response to immune stimulation (PMA) in an immortalized model of latency (Jurkat A1) [248]. These discrepancies may be because of differences in the chromatin landscape at or surrounding the integration site, including the methylation status of nucleosomes encompassing the proviral 5′-LTR and entire genome. While interesting, the disparate results from these studies urge a careful and comprehensive evaluation of all HP1 isoforms in various models of latency in which the role of cell state and integration site placement is carefully explored. Additionally, a thorough investigation is urgently needed to clarify direct and indirect functions (e.g., through chromatin remodeling complex recruitment) of different HMTs and HP1 isoforms in nucleosome positioning at the proviral genome, as well as in the establishment and/or maintenance of proviral latency. Given the large fraction of defective proviruses over intact proviruses [8] and their potential differential locations in the human genome [23], it is yet completely unclear how the various HMTs and HP1 isoforms target and regulate proviral transcription and fate from these disparate physical and functional proviral groups.

Another histone methylation (H3K27me3) has long been known to have a repressive role (reviewed in [249]) in transcription programs regulating key biological outcomes such as differentiation and development [250,251]. Expectedly, H3K27me3 was detected at the proviral genome in the basal state of immortalized models of latency (Jurkat) and further, H3K27me3 alongside H3K9me3 were lost during the transition to the host-viral phases of the program in response to stimulation (TNF-α) [238].

Consistent with the deposition of H3K27me3 at the proviral genome in immortalized models of latency (Jurkat), EZH2 (the catalytic subunit of the polycomb repressive complex (PRC2) responsible for H3K27 di- and tri-methylation) was also found in the basal state; however, its levels rapidly decreased during the transition to the host-viral phases of the program in response to stimulation (TNF-α) [252], in agreement with its known repressive role. Consistently, silencing or pharmacologic inhibition of EZH2 reactivated latent proviruses [252], presumably due to reduced H3K27me3 levels at the proviral genome. As many other transcriptional and chromatin regulators do, EZH2 cooperates with another HMT (EHMT2: euchromatic histone lysine N-methyltransferase 2) to establish and maintain proviral silencing in primary models of latency (CD4^+^ T helper (Th_17_)) and in resting memory T cells isolated from aviremic patients [253].

Together, these initial observations indicate that site-specific H3 Lys methylation (both H3K9me3 and H3K27me3) contributes to proviral latency, potentially by maintaining nucleosomes in a repressed state at the 5′-LTR. Although both H3K9me2/3 and H3K27me3 are involved in gene silencing and can act in a concerted manner for the establishment of heterochromatin loci [249], the functional crosstalk between methylation states in the establishment and maintenance of provirus latency is not fully understood.

More recently, another HMT (SMYD2: SET and MYND domain-containing protein-2) was identified using an RNAi-based screening approach [254]. SMYD2 binds to the proviral 5′-LTR and maintains proviral latency via H4K20me1. Thus, loss or inactivation of SMYD2 function induced latency reversal from both immortalized (Jurkat) and primary (CD4^+^ T) models of latency [254]. Given the site-specific H4 (H4K20me1) and H3 (H3K9me2/3 and H3K27me3) methylation at the proviral genome, elucidating the potential interplay and crosstalk between their chromatin writers, readers, and erasers seems (Figure 4C) an interesting but poorly studied process.

### 6.5. Proviral Genome DNA Methylation

Besides histone PTMs, DNA methylation of the proviral promoter is another epigenetic silencing mechanism implicated in transcriptional regulation. Two CpG islands (at positions ‒194 to ‒94 and +180 to +368 respective to TSS) flanking the proviral TSS in immortalized (Jurkat) and primary (CD4^+^ T) models of latency were proposed [42]. One of those CpG islands was found to be targeted by host methyl–CpG-binding domain protein (MBD2) in both immortalized (Jurkat) and primary (CD4^+^ T) models of latency, thereby promoting latency establishment [42]. Additionally, hypermethylation of the 5′-LTR in latently infected patient CD4^+^ T cells under suppressive therapy conferred resistance to provirus reactivation in contrast to hypomethylated HIV 5′-LTR in viremic patients [255], suggesting that the state of DNA methylation of the proviral promoter correlates with its transcriptional activity. As such, methylation of the proviral genome has been proposed as an additional restriction factor for latency stability but not for the initial establishment of proviral silencing. However, in contrast to these two studies, several others demonstrated very low levels of DNA methylation within the 5′-LTR of latently infected patient CD4^+^ T cells under suppressive therapy [8,256,257], signifying DNA methylation is not the lion’s share in proviral transcription silencing. Collectively, these contrasting results could be due to the fact that a vast majority of latently infected resting CD4^+^ T cells harbor defective proviruses [8] and may contain hypermethylated HIV 5′-LTRs [256].

### 6.6. Conclusions and Future Directions

Recent progress made for the identification and isolation of latently infected CD4^+^ T cells carrying intact proviruses from aviremic patients [8] may clarify and shed light into the role of CpG methylation in promoting and/or maintaining proviral latency. Newer high-throughput sequencing approaches such as NOMe-seq (nucleosome occupancy and methylome sequencing), and more recent single-cell implementations (scNOMe-seq) have been used to obtain locus-specific information of chromatin accessibility, nucleosome phasing, and DNA methylation at the single molecule level [258,259]. Importantly, these newer technologies, especially scNOMe-seq, may provide combinatorial epigenomic signatures of single-cells from heterogeneous populations of patient samples (Figure 2).

Collectively, studies over the past three decades focused on the identification and characterization of host cell factors hijacked by HIV to integrate and transcribe the proviral genome. This was done using either minimalistic models of latency in immortalized clones of CD4^+^ T cells or population-based studies in ex vivo primary models of latency and aviremic patient samples. These previous studies led to the characterization of a vast number of factors with nucleic-acid binding properties and chromatin-modifying or -remodeling activity, just to name a few. However, we currently have a very little understanding on how these chromatin regulatory factors regulate the most important reservoir in patients, namely intact proviruses that are capable of reseeding the reservoir upon therapy interruption. With the advent of single-cell genomic technologies for ChIP-seq [260], ATAC-seq [261], and Hi-C [262] to analyze individual proviruses (reviewed in [263]), and next-generation tools that could simultaneously assess proviral integration sites, intactness, expression and/or maps of active Pol II (Figure 5), a leap in our future understanding of proviral integration and transcriptional regulation can be made.

Most of our understanding of nucleosome positioning relies on data collected in vitro using clones of proviruses in immortalized models of latency (Jurkat) and low-resolution tools, compared to chemical cleavage techniques, such as MNase digestion followed by foot printing analysis. Despite these pioneer discoveries, it remains unclear how applicable these early models are for intact proviruses persisting in aviremic patients. Is the current dogma in the field concerning nucleosome positioning still valid for intact proviruses in patients and/or for models that can better interrogate and explain disease progression? How does the integration site influence the position of nucleosomes in both defective and intact proviruses? How does the orientation and direction of proviruses respective to nearby regulatory elements in the 1D and 3D dimensions (Figure 3) influence the position of nucleosomes in both defective and intact proviruses? How does the integration landscape in both immortalized and primary models of latency in CD4^+^ T cells as well as in patient samples regulate nucleosome positioning at the proviral genome? Can the initial results obtained in the above immortalized models be informative of the physiologic mechanism of chromatin control at integrated proviruses? Are those results generalizable? What is/are the role/s of the integration site, nuclear topology, sub-compartments and chromatin states to histone methylation, nucleosome positioning and proviral fate? Does the precise position and magnitude of H3 and H4 site-specific (or combinatorial) methylation matter in terms of restricting, and or allowing future, proviral activation and latency-reversal potential? These are some of the many questions that remain to be answered to fully understand how histone site-specific PTMs shape proviral responses. Certainly, we face an exceeding limitation due to the scarcity of, not only the latent reservoir but importantly, the “intact” latent reservoir (Figure 5A). Together these limitations hamper the need to interrogate the role of nucleosome positioning and epigenomics in physiologically and clinically relevant models of disease.

## 7. Disease Relevance and Current Therapeutic Challenges

Early on, it has been noted that HIV persistence was due to viral latency and not drug failure citing limited mutation rates [264]. Where latency is maintained by memory CD4^+^ T cell survival (possibly leading to clonal expansion) and homeostatic proliferation ([29], reviewed in [265]), clinical interventions must include targeting not only viral replication, but the rebound provirus that perpetuates in proliferating memory CD4^+^ T cells too ([29], reviewed in [265]). As such, many therapeutic efforts have been initiated which fall under different strategic categories: Reducing the reservoir, immunologically control viral rebound, and silencing the reservoir (reviewed in [265]). It is worth noting these efforts are focused to target the intact inducible provirus, which some may argue is not a complete cure as defective provirus may still express viral proteins able to inflict harm [266]. Nonetheless, to highlight a few strategies, one method for reducing the reservoir is “shock and kill” which boosts the provirus out of latency using latency reversing agents (LRA) so that enhanced cytotoxic lymphocytes may completely purge the virus (reviewed in [267,268]). Immunotherapy includes the use of broadly neutralizing antibodies [269,270] and chimeric antigen receptor T (CAR-T) cells [271] to target viral envelope on the cell surface for killing. A “block and lock” strategy could be used for silencing the reservoir in where pharmacologic inhibitors are used to implore HIV into a permanent transcriptionally silent state [272]. Importantly, the efficiency of these strategies and the underlying molecular mechanisms is often biased to singular latency models used, such that when tested in non-homogeneous patient samples (Figure 2), the efficacy in reducing overall reservoir size is often questioned/limited (reviewed in [2]). Therefore, new attempts to capture generalizable model(s) of latency, such as combining cell line data with ex vivo approaches [236], is often at the forefront of identifying and confirming new compounds. It is clear the varying transcription activity within a reservoir is influenced by the heterogeneous integrated nature within a proviral population [75] (Figure 2). Even so, we must also consider the added complexity stemming from the different origins, or genesis, of each unique proviral integrations within the same individual (Figure 1). Distinguishing how the origin of establishing latency, for example an intact provirus arising from clonal expansion vs uniquely integrated intact proviruses, influences viral rebound and reseeding of active infection upon ART cessation may illuminate the strategic development of novel compounds or combination of treatments.

The importance of using primary models of latency is highlighted in the identification of benzotriazoles as a class of LRAs that inhibit STAT5 SUMOylation increasing its presence on the HIV proviral promoter [273]. Yet, as opposed to primary cells and ex vivo models of latency, benzotriazoles fail to reactivate in multiple immortalized models of latency (J-Lat clones) because they lack the IL-2 receptor (CD25), thereby impeding STAT5 activation [273]. Physiologically relevant models are increasingly being used in combination and/or to verify findings derived from immortalized models. This combination of approaches has aided the understanding of the chromatin landscape and offered insights into predicting and validating long-term efficient new compounds. For example, didehydro-cortistatin A (dCA) was shown to inhibit Tat function by preventing TAR binding to reduce viral replication in vitro using immortalized models of latency (HeLa-CD4-LTR-Luc) and later confirmed using patient samples [274,275]. Yet, even now, we are beginning to bypass immortalized models for mechanistic studies all together. For example, to understand the mechanism of dCA, Valente and colleagues assessed how dCA promotes epigenetic silencing by increasing nuc-1 occupancy at the proviral genome (Figure 4), thereby preventing Pol II from binding the promoter using patient samples ex vivo. These results further confirm long term efficacy using the bone marrow-liver-thymus (BLT) mouse model of latency and persistence [272].

Given the importance of chromatin-modifying complexes in silencing proviral transcription, they were the focus of many studies aimed at promoting latency-reversal and reduction of the latent reservoir. The SWI/SNF chromatin modifying complexes have been targets for the development of therapies in the clinical setting. However, given BAF and PBAF complexes share several subunits and have functionally opposed roles in proviral transcription, this has represented a roadblock for “shock and kill” approaches of an HIV cure. Further, several compounds were recently identified to target subunits of the SWI/SNF BAF complex, but were later reported to have toxic off-target effects [208]. Recently, Dykhuizen and colleagues used a high throughput approach to identify non-toxic small molecule inhibitors of the BAF complex, which led to the identification of macrolactam compounds as LRAs that reverse latency both in immortalized and primary models of latency as well as in CD4^+^ T cells from aviremic patients without causing toxicity or T cell activation [276]. These new compounds target the BAF-specific ARID1A subunit, thereby reducing repressive nucleosome occupancy at the 5′-LTR (Figure 4).

Beyond high-throughput approaches, another tactic others have taken is to provide a better kinetic understanding of the proviral chromatin environment as it transitioned into a latent state in a primary model of latency (CD4^+^ T cells transduced with the with anti-apoptotic molecule Bcl-2) during EMT (Figure 1) [277]. Expectedly, heterochromatin (named based on H3K9me3 or H3K27me3 densities at the provirus (Figure 4)) gradually stabilized as cells transitioned from the active to the resting state, and thus proviruses became less accessible with reduced activation potential. Such a tool may eventually help specify drug development to accurate chromatin environment profiles.

The strongest indications of LRA’s efficacy are in vivo studies using animal models. Garcia and colleagues showed how treatment with the compound AZD5582 in HIV infected BLT mice and SIV infected rhesus macaques under ART induced HIV and SIV RNA expression in several tissues including lymph node of the macaques and lymph node, thymus, bone marrow, liver, and lung of BLT mice [278]. Though no reduction in reservoir size was detected, these models can lead the way to determine dose, timing and pairing with other compounds to induce the cytotoxic lymphocyte killing of the persistent reservoir. As shown by Silvestri and colleagues though the IL-15 super agonist compound, N-803, is able to reactivate primary human CD4^+^ T cells latently infected with HIV in vitro, co-culturing with CD8^+^ T cells inhibited N-803–mediated latent provirus reactivation [279]. This was later confirmed in SIV infected macaques, where N-803 alone had no impact on the reservoir. Yet, combining N-803 with MT807R1, an antibody that depletes CD8^+^ T cells in vivo resulted in a robust SIV increase. Together, these studies illuminate how animal models can be used not only to test compound efficacy but also to develop latency reversing strategies that take into account physiologic variables.

Overall, the importance of cross-validation using different models of latency, with emphasis on physiologically relevant models, is becoming increasingly apparent. Additionally, a careful analysis of hundreds or even thousands of single proviruses integrated in unique positions and bearing unique genetic variants, and not population-based studies, will hone our understanding on the role of integration landscape and cell state on latency reversal potential and other therapeutic schemes.

## 8. The Ephemeral Nature of Ideas and Considerations for Future Research

Characterizing the latent reservoir composition was the first core discovery spurring an entire field dedicated to understanding the mechanisms underlying its establishment and maintenance. Though initially capturing the precise reservoir size and composition were low resolution estimates (i.e., either over- or under-estimating the size and intactness), improved and combined methodologies have continuously revised the latent reservoir profile, unveiling that only a miniscule proportion is intact and inducible (Figure 5A). By surveying the genetic environmental landscape in where proviruses have embedded themselves, many were able to define peculiar preferential “behaviors” of this intriguing virus. RIGs, genes where the provirus was found repeatedly integrated in patient samples, began to reveal the integration site could largely influence what would become the position effect phenomena (Figure 3) [75], in where the integration site neighborhood shift proviral activity. Indeed, it is clear the partialities extend beyond the 1D view of simply a proviral insertion into a linear scale, as the importance of 3D nuclear architecture and temporal order of events in respect to cell state have become more apparent. Nonetheless, each new uncovered characteristic adds to understanding the proviral integration code, or molecular rules tempering proviral fate. At the same time, there is a growing consensus that many of these preferences might only be observed in patient samples and lost in immortalized and/or primary models of latency suggesting limitations in applicability of models potentially attributed to in vivo “selective pressures” and fitness of select proviral groups not recapitulated in vitro.

Beyond fixed provirus physical location and compartmentalization, other influences of latency establishment and maintenance are much more fluid. Cell status, as illustrated by large fluctuations of nuclear transcriptional regulators and coregulators, can galvanize transcriptional responses when transitioning from resting to active status. The factor fluctuations may also be small enough only to generate heterogenized threshold within a state (at rest for example) due to stochasticity. Or the fluctuations may be at the local gene level where deposition of epigenetic marks, Pol II pausing, and/or nucleosome positioning can influence the inducibility of an otherwise intact provirus. Again, many conflicting results are often attributed to differences in models used for their respective studies given the multiple layers of variability (Figure 1B).

Because of the multifactorial aspects contributing to latency (establishment, maintenance, and reversal), it is obvious the era of singular cause is over. From a therapeutic perspective, the failure of LRAs has always been attributed to the same reason: inability of a single drug, which often target a single molecular rule of latency (e.g., HDAC inhibitor, BRD4 inhibitor, etc.), to reactivate the entire latent reservoir. This extends to screening for new drug compounds. In the “new era”, it will be imperative to integrate multiple approaches (such as multiple genetic datasets, epigenetics, transcriptomes, proteomics, 3D genome architecture, nuclear topology, chromatin states, transcriptional activity, proviral position, and intactness) followed by deep learning [75] to predict and experiment with emphasis at the single-cell level and across multiple models including patient samples, and interrogate “deep” (Figure 5B). Understanding how each molecular rule fits together (simultaneously or stepwise) and which rules govern individual proviruses (i.e., is it the same set of rules for all proviruses or are there different kits/sets of rules for different proviruses) will aid personalized therapeutic development and eventual eradication of HIV. However, one obvious but extreme challenge is how the dataset integration (Figure 5B) will be achieved if patient samples are composed of a scarce and heterogeneous population in which multiple CD4^+^ T cells contain proviruses integrated in unique regions in addition to the clonally expanded pool.

As physiological relevance is vital to remove distracting or inapplicable models/mechanisms from the discussion, preference must be given to use patient samples or primary cell systems. Yet, with the limitation of the number of intact proviruses within a patient to collect data from, alternative models may need to be developed. In this context, can primary models of latency be created in the presence of ART, without the ectopic expression of anti-apoptotic factors but allowing their long-term culturing without causing T cell exhaustion? With the advent of genome editing tools [280], can we engineer “anatomically” correct in vitro models of latency (in immortalized but noncancerous cells or naïve healthy donor cells) in where proviruses are directed to the same integration sites as those found in patient samples? Would this approach potentially remove the roadblock of limited number of cells? No doubt, this would raise additional questions as it would uncouple the specific integration site from latency status, i.e., would repeating the same integration site (not by clonal expansion, but by executing into the same insertion site) repeatedly yield the same “inducibility” or not? Will the integration site landscape remain the same? If the results are consistent, then the molecular rules must be stable and therefore targetable.

With our efforts to discuss a surmountable number of key discoveries and situate them in the context of previously published literature, we hope to provoke future work that will advance our understanding of this fascinating biomedical research challenge. New data supporting or refuting the models discussed herein will undoubtedly increase our knowledge and generate discussions to help with future clinical applications rather than engendering unproductive controversy. Even the more skeptical researchers should be attracted to challenge these models and bring their own points of view to fuel progress in the field.

## Figures and Tables

**Figure 1 viruses-12-00555-f001:**
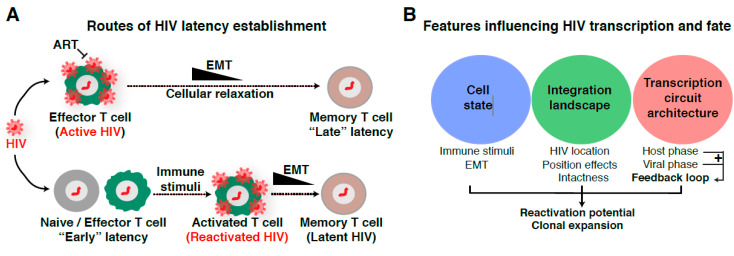
Routes of latency establishment and features influencing HIV proviral transcription and fate. (**A**) In “early” latency, HIV integrates into genomic domains not supporting transcription, but is otherwise intact and induced by immune stimuli. Alternatively, HIV infects cells in different states of activation or during effector-memory transition (EMT) [35,36] in which cellular relaxation pushes an initial transcriptionally active provirus into a silent provirus promoting “late” latency. (**B**) List of features contributing to HIV proviral transcription and fate. The “cell state” denotes CD4^+^ T cells of any of the major subsets (e.g., Th1, Th2, Th17, Treg, Tcm) in different stages (naïve, effector, resting memory, and effector memory). The “integration landscape” describes the heterogeneous positions of HIV proviruses in the human genome and position effects. The complex “transcription circuit architecture” describes the progressions through the different phases of the HIV transcriptional program (basal, host, and viral) leading to the positive-feedback loop. See text and Figure 4A for details.

**Figure 2 viruses-12-00555-f002:**
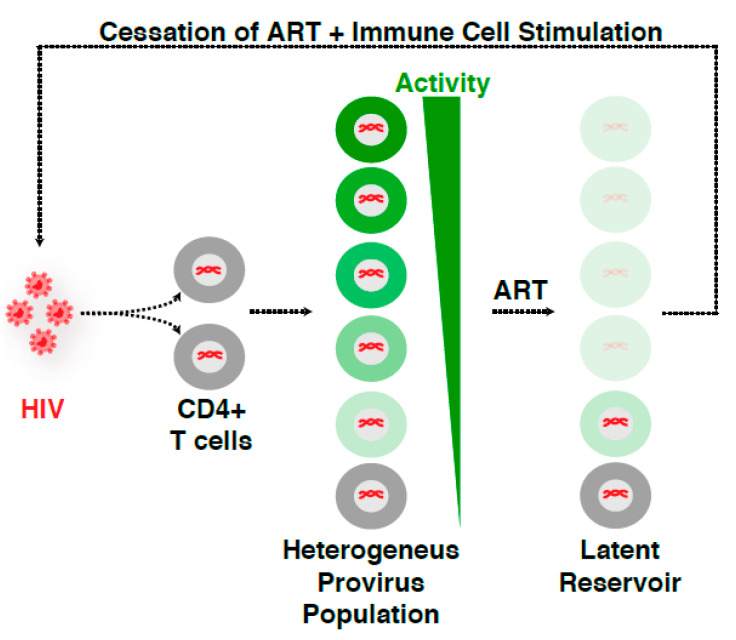
HIV infection leads to a heterogeneous proviral population. HIV infection of CD4^+^ T cells yields a “heterogeneous provirus population” with a variable, continuum degree of gene expression and replication. ART stops viral replication and disease progression into the AIDS phase. However, ART is unable to cope with both the transcriptionally silent proviruses and those replicating at very low levels, thereby surviving therapy and generating the so-called “latent reservoir”. As such, ART discontinuation in the presence of immune stimulation results in a rapid rebound of virus, indicating that while ART suppresses viral replication, HIV is able to persist in an infectious state for years.

**Figure 3 viruses-12-00555-f003:**
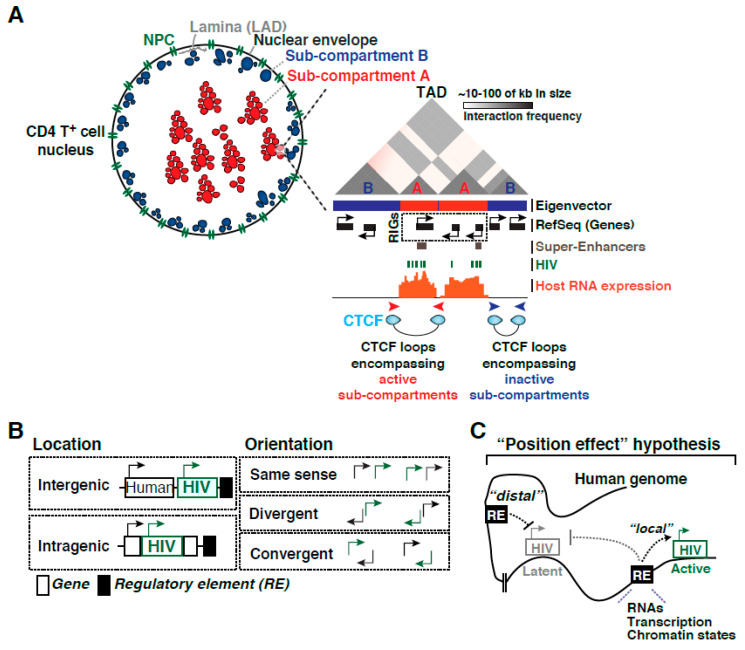
Contribution of nuclear topology shaping the integration landscape, transcriptional regulation, and proviral fate. (**A**) Scheme depicting the relationship between nuclear topology, nuclear envelope, nuclear lamina with its associated domains (LADs), topological associated domains (TADs), the position of the eigenvector respective to nuclear sub-compartments A and B, and their genetic composition (genes and regulatory elements like super-enhancers) with their activity (host RNA expression). The position of RIGs in the RefSeq track and other chromatin accessible regions of the human genome “preferred” for HIV integration (such as super-enhancers) are highlighted. (**B**) Summary of HIV insertion patterns (both location and orientation) relative to genes and regulatory elements in the human genome (e.g., enhancers, mobile elements). (**C**) Position effect hypothesis highlighting both local and distal effects from human genome regulatory elements and long-range chromatin interactions, thereby influencing proviral fate.

**Figure 4 viruses-12-00555-f004:**
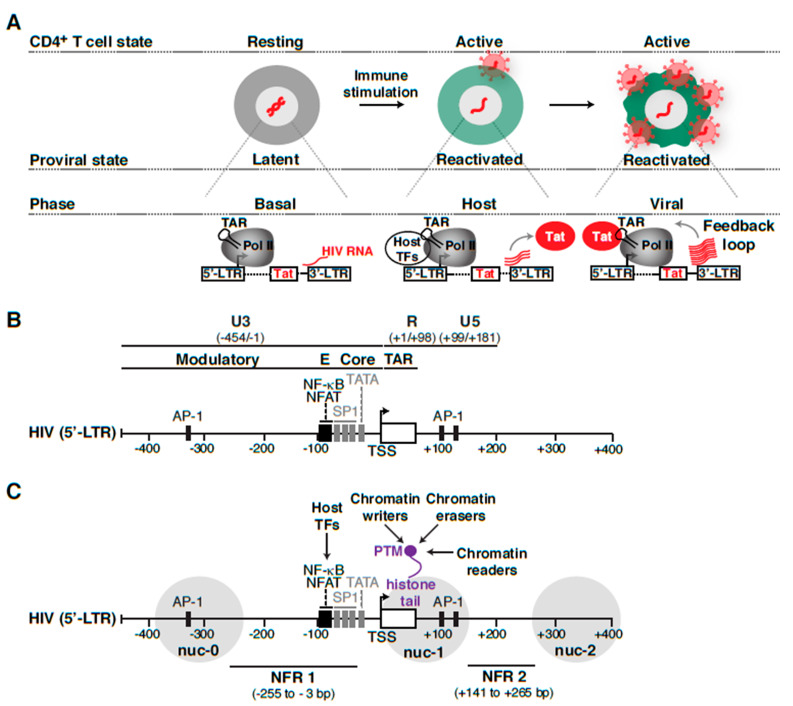
The multiple layers of control in the HIV transcriptional program. (**A**) Schematic representation of the progressions through the HIV transcriptional program. See text for details. (**B**) Schematic representation of the HIV 5′-LTR showing U3, R, and U5 regions. The U3 region is further divided into the modulatory region, enhancer (E), core promoter, and TAR. The U3 and R regions comprise binding sites for SP1, NF-κb/ NFAT, and AP-1 transcription factors. (**C**) A depiction of chromatin landscape at the HIV promoter showing the three well-positioned nucleosomes (nuc-0, nuc-1, and nuc-2), separated by two large NFRs (NFR-1 and NFR-2) in the basal state (latent proviral fate). The action of chromatin modifiers (Readers, Writers, and Erasers) render the position of nuc-1 downstream of the TSS and regulate the establishment of proviral latency. The position of one histone tail and a general PTM (i.e., acetylation and methylation) is shown as a simple representation. See text for details.

**Figure 5 viruses-12-00555-f005:**
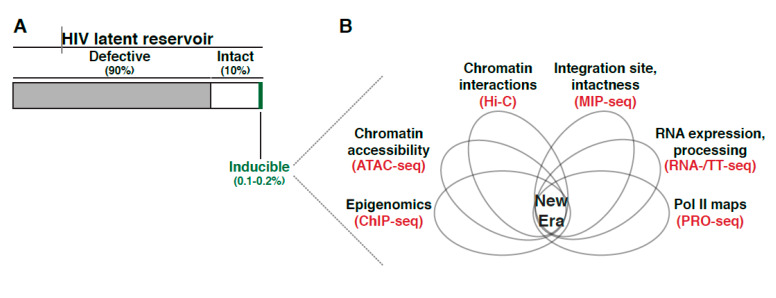
Studies of the latent reservoir in the “new era”. (**A**) The latent HIV reservoir is comprised of a large fraction of proviruses with replication-defective genomes (~90%) and a small fraction of proviruses with replication-intact genomes (~10%). From the intact pool, only a tiny (~0.1–0.2%) fraction can be induced after single or multiple rounds of stimulation with potent T cell agonists ex vivo. (**B**) The “new era” will require use of newer omics tools implemented at the single-cell level allowing for unprecedented opportunities to investigate the multiple facets of the latent reservoir (including integration site, intactness, expression, chromatin accessibility/nucleosome positioning, and mechanistic regulation) for both, patient samples and ex vivo primary models of latency. The wealth of information generated by each of these technologies, integrated using deep learning algorithms, would be akin to a “Swiss army knife” in where any tool or combination of tools could dissect the multifactorial aspects contributing to proviral latency.

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
