# Peer review of "HIV-1 Proviral Transcription and Latency in the New Era"

_viruses, 2020, doi:10.3390/v12050555_

Round 1

Reviewer 1 Report

This is a well-written and comprehensive summary of knowledge regarding HIV transcription and latency. The appropriate literature is cited and there’s a nice balance of literature summary and speculation.  I have only a couple of minor suggestions:

1) Work from the Weinberger lab on the role of stochasticity in HIV transcription is only mentioned in passing, and probably deserves a little more discussion.

2) Discussion about the so-called “new era” seems somewhat vague and unscientific. Continuous technological changes means we are always entering a “new era” – its not clear to me how the way this term used is really specific or useful.

3) There are a number of typos and grammatical errors that should be fixed.

Eg:

Line 147 – “provirus” should be “proviruses”

Line 324 – should be “AND is controlled by”

line 338 – multiple “years “of control? Should be “multiple levels”?

Line 449 – “subsequence” phases should be “subsequent phases”

Line 526 –should be  “role of integration in the control of basal transcription”?

Line 961 –should be “boosts the provirus out of latency”?

Line 1039  “with high resolution definition”?

Author Response

Point-by-point responses to the Reviewer’s comments

Reviewer #1

This is a well-written and comprehensive summary of knowledge regarding HIV transcription and latency. The appropriate literature is cited and there’s a nice balance of literature summary and speculation.  I have only a couple of minor suggestions:

We thank the reviewer for her/hist constructive criticisms which will help us improve our manuscript. Below we answer to each specific suggestion/request.

Point 1. Work from the Weinberger lab on the role of stochasticity in HIV transcription is only mentioned in passing, and probably deserves a little more discussion.

We totally agree with the reviewer. To answer this suggestion, we have expanded on the role of stochasticity in HIV transcription and have incorporated the required references. This is now discussed (pg. 10, lines 403-437).

Point 2. Discussion about the so-called “new era” seems somewhat vague and unscientific. Continuous technological changes means we are always entering a “new era” – its not clear to me how the way this term used is really specific or useful.

We completely agree with the reviewer that the description of “new era” was vague so we have clarified how this term is used in the Abstract and throughout the text. We personally think that introducing this concept and its explanation earlier and throughout the text (e.g., pg. 1, lines 18-20; pg. 4, lines 155-162; pg. 8, line 344; pg. 25, line 1150) now better helps the reader navigate through a discussion of previous discoveries and how implementation of future approaches in this “new era” will benefit studies of proviral transcription and latency. We also agree with the fact that continuous technological changes facilitate entering new eras. However, our use of this term is now precisely explained and does not rule out the introduction of future technological advances. Surprisingly, a quick Pubmed search retrieved more ~16,000 studies using this term in their titles so strongly feel like with the new additions/explanations the rationale of its use is better justified.

Point 3. There are a number of typos and grammatical errors that should be fixed.

We have carefully revised the grammar and fixed a number of typos and errors as indicated below.

Eg:

Line 147 – “provirus” should be “proviruses”

This is now fixed (now line 154)

Line 324 – should be “AND is controlled by”

This is now fixed (now line 359)

line 338 – multiple “years “of control? Should be “multiple levels”?

This is now fixed (Figure 3 legend)

Line 449 – “subsequence” phases should be “subsequent phases”

This is now fixed (now line 521)

Line 526 –should be “role of integration in the control of basal transcription”?

This is now fixed (now line 601)

Line 961 –should be “boosts the provirus out of latency”?

This is now fixed (now line 1044)

Line 1039  “with high resolution definition”?

This is now fixed (sentence removed in line 1123)

Reviewer 2 Report

The review focuses and summarizes aspects related to proviral transcription and HIV latency. It provides a detailed description of background knowledge and experimental findings and is structured into several parts: first, review of features related to the HIV latent reservoir; second, review of transcriptional regulation of the provirus in different conditions; third, review of epigenetic aspects of regulation of proviral transcription and latency; fourth, review of how findings on proviral transcriptional regulation have been translated into the clinic/clinical trials; fifth: considerations for future research in the field.

The main strength of the review are parts two and three, namely the sections reviewing mechanistic aspects of proviral transcription (line 323 to 658) and mechanisms of epigenetic regulation thereof (line 659 to 901). These sections are well-structured into subsections (although subsections should be made more clearly for the part on the transcriptional program) and provide a well readable overview of the field with adequate detail, which is not often covered in reviews in the HIV latency community.

Sections four (line 950 to 1032) and five (line 1033 to 1098) are certainly justified. They give a glimpse into bench-to-bedside translation of HIV latency research and an outlook on potential future research directions in view of the multitude of aspects that might play a role in HIV latency. The paragraph on single cell techniques (line 902 to 914 and Fig 5) and the paragraph on open research questions (line 927 to line 949) should however be added to section five. They are out of context in the epigenomics part.

The main weakness of the review as it stands is the first part on different aspects and phenomena related to the HIV reservoir (line 21 to 322). This section is not well structured and makes for a rather difficult and unclear read. It touches on aspects that have been reviewed in a better and more detailed way elsewhere. Although this section is required as an introduction for the subsequent parts of the review, it should be kept much shorter and more focused and refer to other reviews. This would also shorten the review in general, which might be a good idea in view of the current length.

Further points to address:

  1. the term ‘new era’ in the title and in the manuscript (eg. line 166, 655 etc.) is not explained – why do the authors see a new era in latency research? What defines this era? This should be clarifies otherwise the term should not be used
  2. Certain paragraphs seem too detailed and diverge from the central topic of the review. They should be left out or shortened – also in view of the current length of the manuscript. Eg: lines 227 to 248 (description of nuclear compartments).
  3. Fig 3A is not very informative and appears too detailed for purpose of this review.
  4. The title of Fig4 is not comprehensive – ‘multiple years??? of control’….
  5. language points:
  • terms like ‘great’ (eg line 174, 285 etc) ‘famous’ (eg line 27, 119 etc) are very subjective and should be avoided.
  • Line 33: replace ‘most well’ by ‘best’
  • The authors often refer to the fact that a theme is taken up later in the review. This is generally not necessary and sounds quite repetitive if used too often (eg. line 727, 787)

Author Response

Point-by-point responses to the Reviewer’s comments

Reviewer #2

The review focuses and summarizes aspects related to proviral transcription and HIV latency. It provides a detailed description of background knowledge and experimental findings and is structured into several parts: first, review of features related to the HIV latent reservoir; second, review of transcriptional regulation of the provirus in different conditions; third, review of epigenetic aspects of regulation of proviral transcription and latency; fourth, review of how findings on proviral transcriptional regulation have been translated into the clinic/clinical trials; fifth: considerations for future research in the field.

Point 1. The main strength of the review are parts two and three, namely the sections reviewing mechanistic aspects of proviral transcription (line 323 to 658) and mechanisms of epigenetic regulation thereof (line 659 to 901). These sections are well-structured into subsections (although subsections should be made more clearly for the part on the transcriptional program) and provide a well readable overview of the field with adequate detail, which is not often covered in reviews in the HIV latency community. Sections four (line 950 to 1032) and five (line 1033 to 1098) are certainly justified. They give a glimpse into bench-to-bedside translation of HIV latency research and an outlook on potential future research directions in view of the multitude of aspects that might play a role in HIV latency. The paragraph on single cell techniques (line 902 to 914 and Fig 5) and the paragraph on open research questions (line 927 to line 949) should however be added to section five. They are out of context in the epigenomics part.

We thank the reviewer for the time spent evaluating our manuscript, the positive remarks and for a number of important constructive criticisms. Below we discuss how we used the reviewer’s comments to strengthen our work.First, we agree with the reviewer that adding subsections to the transcriptional program in section 3 will improve its presentation. This is now indicated (pg. 9-14) as well as throughout the review for consistency. Second, we agree that the paragraphs dedicated to single cell approaches and open research questions are part of the final discussion section. However, the single cell approaches discussing chromatin and epigenetics belong to the “epigenomics section” because they are used to address this problem.

Point 2. The main weakness of the review as it stands is the first part on different aspects and phenomena related to the HIV reservoir (line 21 to 322). This section is not well structured and makes for a rather difficult and unclear read. It touches on aspects that have been reviewed in a better and more detailed way elsewhere. Although this section is required as an introduction for the subsequent parts of the review, it should be kept much shorter and more focused and refer to other reviews. This would also shorten the review in general, which might be a good idea in view of the current length.

We partially agree with the reviewer regarding the usefulness of section 1 because Reviewer #1 fully appreciated it and because it is strictly needed to better guide the reader to the subsequent parts of the text. Section 1 and its associated figures (Figure 1 and 2) are used throughout the entire text so deleting this section will make the presentation of the story impossible. We feel like our review brings together the best of different worlds (namely, the latency problem by first introducing the latent reservoir, and then focusing on the transcription regulatory mechanisms to understand the latent reservoir). However, as requested by the reviewer we have reduced its length by eliminating unnecessary parts and by referencing other reviews when appropriate (pg. 1-7). Overall, we feel that these changes have improved readability of section 1, reducing the overall length of the manuscript, although there were no character limits per our conversation with the journal’s Editors.

Further points to address:

Point 3. the term ‘new era’ in the title and in the manuscript (eg. line 166, 655 etc.) is not explained – why do the authors see a new era in latency research? What defines this era? This should be clarifies otherwise the term should not be used

We agree with the reviewer that our description of “new era” was lacking clarity. We have now updated the manuscript introducing the term early in the abstract, introductory section and throughout the text to explain how the previous discoveries and up-to-date technologies will lead us to a “new era” of better understanding the proviral transcription and latency. We also emphasize the “new era” includes the not-yet-thought-of future development of newer and/or improvement of current technologies (e.g., pg. 1, lines 18-20; pg. 4, lines 155-162; pg. 8, line 344; pg. 25, line 1150).

Point 4. Certain paragraphs seem too detailed and diverge from the central topic of the review. They should be left out or shortened – also in view of the current length of the manuscript. Eg: lines 227 to 248 (description of nuclear compartments).

This is a good suggestion. We have now removed unnecessary details regarding nuclear sub-compartments (see deleted sections in pg. 7, lines 257-271).

Point 5. Fig 3A is not very informative and appears too detailed for purpose of this review.

The title of Fig 4 is not comprehensive – ‘multiple years??? of control’….

4A. We agree with the reviewer that Figure 3A needed more details and a better relationship to HIV proviral transcription and latency. We have now provided more clarifications in the text (see Figure 3A legend and pg. 5-7) to allow readers better understand this section.

4B. We apologize for the oversight in the Figure 4 title, which should read “multiple layers”. This is now fixed (Figure 4 legend, now line 373).

Point 6. language points:

5A. Terms like ‘great’ (eg line 174, 285 etc) ‘famous’ (eg line 27, 119 etc) are very subjective and should be avoided.

These are good points and have now been fixed throughout the text.

5B. Line 33: replace ‘most well’ by ‘best’

This is now fixed (now line 34)

5C. The authors often refer to the fact that a theme is taken up later in the review. This is generally not necessary and sounds quite repetitive if used too often (eg. line 727, 787)

This is a good point and has now been fixed when required (eg., lines 44-45).

Round 2

Reviewer 2 Report

The authors have recognized and addressed all comments by the reviewer. The revised manuscript has definitely profited from the review. It now has more clarity and reads much better.

In my opinion, the manuscript now qualifies for publication in Viruses.